# How Method Matters: The Impact of Material Characterisation Techniques on Liquid Silicone Rubber Injection Moulding Simulations

**DOI:** 10.3390/polym17223086

**Published:** 2025-11-20

**Authors:** Maurício Azevedo, Silvester Bolka, Clemens Holzer

**Affiliations:** 1Polymer Competence Center Leoben GmbH, 8700 Leoben, Austria; mauricio.azevedo@pccl.at; 2Faculty of Polymer Technology, 2380 Slovenj Gradec, Slovenia; silvester.bolka@ftpo.eu; 3Institute of Polymer Processing, Department Polymer Engineering and Science, Montanuniversität Leoben, 8700 Leoben, Austria

**Keywords:** liquid silicone rubber, injection moulding, curing kinetics, specific heat capacity, thermal conductivity, specific volume, processing simulation accuracy

## Abstract

Injection moulding of liquid silicone rubber (LSR) requires reliable computer-aided engineering simulations to support process optimisation, which in turn depend on accurate material data. In this study, thermo-physical and kinetic properties of a highly filled injection moulding (IM) grade of LSR were systematically characterised using complementary experimental approaches, and their impact on simulation fidelity was critically assessed. Specific heat capacity was measured using both modulated DSC and the standard sapphire method, revealing temperature dependence but no intrinsic change during curing, with sapphire-based data incorporating enthalpic effects more realistically for process prediction. Thermal conductivity was found to be nearly constant across the processing temperature range. Curing kinetics were investigated by calorimetry and rheology, with the former supporting an autocatalytic mechanism and the latter suggesting an $n^{th}$-order model, reflecting differences in detection sensitivity and onset characterisation. When implemented into injection moulding simulations, viscosity primarily affected injection pressures, while differences in specific heat capacity and curing kinetics strongly influenced predicted curing profiles and cycle times. These results emphasise that dataset choice, particularly for curing-related parameters, is critical to achieving predictive accuracy in LSR injection moulding simulations. Unlike previous studies on LSR injection moulding, which typically adapt thermoplastic-inspired characterisation methods without systematically addressing their limitations, this work introduces an organised and comparative methodology to evaluate how different material characterisation techniques influence simulation outcomes. The proposed approach establishes a methodological framework that can guide future research and improve the reliability of process simulations for LSR and other polymeric systems.

## 1. Introduction

Process optimisation within the injection moulding context is a complex task considering the intrinsically complicated manufacturing process via IM. As described by Mitsoulis [1], the injection moulding process comprises time-dependent, 3D, compressible, and non-isothermal flows, with moving free boundaries, and, in the case of liquid silicone rubber IM, holding a chemical reaction inside the mould. Such optimisation involves intense knowledge and experience regarding the process itself, being time- and resource-consuming. In addition, considering the widespread employment of injection moulding techniques, minimal marketing time also plays an important role when optimisation is performed, aiming to become more competitive [2]. In this sense, computer-aided engineering (CAE) simulation technologies based on computational fluid dynamics (CFD) are widely employed to optimise the injection moulding process, since they present the following non-exhaustive list of assets [3]:It is cheaper and less time-consuming, since trial-and-error analyses based on industrial pilot lines would be unaffordable.Numerical tools allow the evaluation of a material response without the physical use of the real material and without compromising mould manufacturing.CAE simulation routines provide the possibility to shorten the cycle time and optimise the curing duration to save energy and avoid material waste.Simulation is able to avoid re-design of the mould.CFD studies are able to predict processing-related defects in the injected part, such as weld lines and air traps.

To achieve the mentioned benefits, the performed simulation must be reliable in a sense that it mimics the real material behaviour and, therefore, the real IM process. The accuracy of such a simulation depends on many factors, such as the representation of the mould and cavity geometries, the chosen flow modelling method, the realistic set of runner, sprue, and gating system, and the material data that is provided as input to solve the governing equations of mass, momentum, and energy conservation [4]. If the aim of the simulation is to predict processing parameters, such as pressure and temperature profiles, and cycle time (which includes the curing time for a reactive material injected into the mould’s cavity), precise material data must be provided, since these feed the phenomenological models that are able to explain the fluid mechanics, heat transfer, and chemical reaction kinetics/thermodynamics associated with the IM process.

The injection moulding simulation of LSR has been unsatisfactorily published in the available literature. In 2002, Haberstroh et al. [5] proposed a material data strategy to simulate the filling and curing phases of LSR. Shortly after, the research advanced and Capellmann et al. [6] enhanced a tool for the simulation of the injection moulding process, enabling it to take into account undervolumetric filling. However, when determining the viscosity of LSR, the authors could not prove the validity of the employed approach for low shear rates and high temperatures, applying plate–plate rheometers. Matysiak et al. [2] extended the previous work and included the simulation data from pressure measurements to gather knowledge on thermal expansion of LSR, but with a simple pressure–temperature model. More recently, Ou et al. [7] investigated the simulation of two-component injection moulding of silicone rubber into a thermoplastic polymer. All the cited investigations reached reasonable simulation results when the predicted processing parameters were compared to the measured values. However, most studies adapted material data characterisation techniques originally developed for thermoplastic materials and employed them with several approximations. For example, none of the mentioned authors highlighted the strong influence of filler content on the applicability of the selected methods or discussed which of the available techniques is most appropriate to define the curing behaviour of LSR. Moreover, the literature generally lacks a methodological rationale explaining the selection and limitations of the experimental procedures adopted for thermoset systems.

From the literature, precise guidelines on material data characterisation of LSR, based on a systematic and comparative investigation of the available state-of-the-art methods, were not found in the context of injection moulding simulation. Therefore, despite the positive results previously reported for LSR simulations, a fundamental methodological gap still exists. The present work aims to fill this gap by introducing an organised and comprehensive framework for evaluating and comparing different material characterisation approaches (covering viscosity, thermal, and curing properties) and by demonstrating how these methodological choices influence the accuracy of injection moulding simulations. This systematic perspective is expected to guide future research, improve reproducibility, and support the development of reliable simulation workflows not only for LSR but also for other polymeric systems.

## 2. Materials and Method

Liquid silicone rubber (Silopren^*TM*^ LSR 2070, $M_{w}$ = 86,673 g·mol^−1^, $M_{w}/M_{n}$ = 1.603, hardness after cured = 70 Shore A), containing approximately 32 wt% (high content) of an inorganic filler, was supplied as a 2-component (A and B) system by Momentive Performance Materials Inc. (Niskayuna, NY, USA). LSR characterisation in terms of molecular weight and filler content is described in a previous publication of our group [8]. Silopren^*TM*^ LSR 2070 is a standard liquid silicone rubber for injection moulding processes and has a mixing ratio of components A:B = 1:1. Mixing of components A and B was accomplished with a dual asymmetric centrifuge (DAC 400.2 VAC-P, Hauschild Speed Mixer, Hamm, Germany) at room temperature, under vacuum, and according to the following step-wise procedure:800 rpm, 2 min, 800 mbar vacuum;1200 rpm, 2 min, 400 mbar vacuum;1600 rpm, 2 min, 100 mbar vacuum;1800 rpm, 4 min, 50 mbar vacuum.

Since this work has an intensive methodological effort, all techniques employed for this investigation are explained in detail below, aiming to allow reproduction of the methods. For all experiments, the same material was investigated, either as a single part (A or B) or as a mixture (1:1 A + B).

### 2.1. Determination of the Viscosity

To measure the viscosity of liquid silicone rubber, two distinct methodologies were employed: large-amplitude oscillatory shear (LAOS), a rotational method; and one employing a high-pressure capillary rheometer (HPCR), a pressure-driven-based procedure. These two routines are detailed in a previous publication [8] from our group. These data were employed here to obtain model parameters for the injection moulding simulation. The viscosity models and their parameters are shown in the Section 3.5.1 of this paper.

### 2.2. Determination of the Specific Heat Capacity

The specific heat capacity was studied employing two different but similar methods. While the standard ASTM E1269 [9] employs sapphire as a reference material for $c_{p}$, the temperature-modulated approach is a stand-alone method to determine a polymer’s $c_{p}$. Since these methodologies differ on how $c_{p}$ is determined, they are critically compared. Both methodologies determined $c_{p}$ as a function of temperature for the liquid silicone mixture A + B, considering two heating cycles in the dynamic scanning calorimeter (DSC1 and DSC2 STAR System Mettler-Toledo International Inc., Greifensee, Switzerland).

#### 2.2.1. Sapphire Method ASTM E1296-11

Approximately 10–20 mg of the A + B 1:1 mixture (triplicate) was placed into aluminium crucibles aiming for maximum possible contact with the crucible’s bottom. Subsequently, and according to the standard ASTM E1269 [9], the following thermal program was applied under 50 mL·min^−1^ of nitrogen gas:Isotherm for 4 min at 50 °C.Heating at 20 K·min^−1^ until 200 °C.Isotherm for 4 min at 200 °C.Cooling at 20 K·min^−1^ until 50 °C.Isotherm for 4 min at 50 °C.Heating at 20 K·min^−1^ until 200 °C.Isotherm for 4 min at 200 °C.
The temperature program was designed to measure the sample’s specific heat capacity as non-cured and during curing (step 2), as well as cured (step 6).

The same thermal program was employed for an empty aluminium crucible (reported weight) and for an aluminium crucible containing the specific heat capacity standard (synthetic sapphire disk with reported weight). It is important to highlight that the software connected to the device only automatically calculates the specific heat capacity if the empty pan is tested before all samples/sapphire and is used to subtract the specimen holder’s thermal response from the sample’s/sapphire’s. In this sense, the sample’s $c_{p,s}$ (J·g^−1^·K^−1^) can be calculated in terms of the standard’s $c_{p,st}$ as(1)$$c_{p,s}=c_{p,st}\;\cdot \;\frac{D_{s}\;\cdot \;W_{st}}{D_{st}\;\cdot \;W_{s}}$$

$D_{s}$ is the heat flow difference (mW) at a given temperature between the empty pan and the sample; $D_{st}$ is the heat flow difference (mW) at a given temperature between the sapphire standard and the sample; $W_{s}$ is the sample’s mass (mg); and $W_{st}$ is the sapphire disk’s mass (mg). The sample’s mass was measured before and after the thermal program and any sample that underwent mass loss higher than 0.3% was repeated, as advised by the ASTM E1269 standard.

From Equation (Equation 1), one can realise that any disturbance in the sapphire heat flow signal will impact the sample’s $c_{p}$ measurement, since $c_{p}$(s) is directly calculated from $c_{p}$(st). This feature is not present in the next methodology as it will be described next.

#### 2.2.2. Modulated Temperature Calorimetry

This approach, commonly referred to as MTDSC (modulated temperature DSC), differs from the one described before due to the fact that it applies a sinusoidal thermal perturbation instead of a constant heating rate for non-isothermal experiments [10,11]. In general terms, during a DSC run, the total heat flow $\frac{dQ}{dt}$ is a contribution of one signal that is dependent on $\frac{dT}{dt}$ and another that is dependent on the value of the temperature *T*. If there is no significant temperature gradient in the sample, i.e., the sample is able to increase its temperature as a whole, the total heat flow during a DSC run can be mathematically expressed as(2)$$\frac{dQ}{dt}=c_{p,t}\;\cdot \;\frac{dT}{dt}+f\left(t,T\right)$$
where $c_{p,t}$ is the specific heat capacity at constant pressure defined here as that due to the energy stored as motion of the sample’s molecules; and f(t,T) is a function that governs the kinetic response of temperature-driven transformations, such as crystallisation, melting, glass transition, or crosslinking. Following this context, there are two contributions to the DSC response: one thermodynamically governed by $c_{p,t}$ (here with sub-index *t* to refer to as the thermodynamic $c_{p}$) and is dependent on $\frac{dT}{dt}$, and another that is kinetically hindered by a mechanism (f(t,T)), which is dependent on *T*. In the case of liquid silicone rubber, the first response is connected to the vibrational, rotational, and translational motions of the poly(siloxane) oligomers, while the second response will be governed by LSR’s curing reaction for temperatures above room temperature. One can realise that, as the temperature rises due to a positive $\frac{dT}{dt}$, the sample’s response to the bond breaking and forming process of the cure reaction and the response due to an increase in oligomers’ kinetic energy (molecular motion) are different. This output signal separation between purely thermodynamic and kinetic-governed phenomena is the main difference between conventional (sapphire method) and modulated DSC methods for $c_{p}$ determination.

To accomplish this distinction, the sample is subjected to a modulated thermal program with an initial temperature $T_{0}$, a heating rate *b*, and a modulation factor with amplitude *B* and frequency ω, as follows:(3)$$T=T_{0}+b\;\cdot \;t+B\;\cdot \;sin\left(\omega t\right)$$

For the present research, stochastic temperature modulations, i.e., well-distributed temperature perturbations within the analysed temperature range with random duration, are superimposed on an underlying rate of conventional DSC. This was accomplished by employing a dynamic scanning calorimeter and the TOPEM^*TM*^ software (v.19.0, Mettler-Toledo International Inc., Greifensee, Switzerland). By applying discrete Laplace transformations, this method is able to determine the quasi-static heat capacity or the thermodynamic specific heat capacity. This $c_{p}$ signal will be independent of any thermal event that occurs during the DSC run. The MTDSC-TOPEM experiments were conducted with an underlying heating rate *b* = 2 K·min^−1^, an amplitude *B* = ±0.5 K, and a period ($\frac{1}{\omega }$) = 15–30 s under 20 mL·min^−1^ of nitrogen gas, employing samples in triplicate. The temperature program is as follows:Isotherm for 1 min at 50 °C.Heating at 2 K·min^−1^ until 250 °C with temperature modulation.Isotherm for 1 min at 250 °C.Cooling at 2 Kcmin^−1^ until 50 °C without temperature modulation.Isotherm for 1 min at 50 °C.Heating at 2 K·min^−1^ until 250 °C with modulation.
The temperature program was designed to measure the sample’s specific heat capacity as non-cured and during curing (step 2), as well as cured (step 6). These experimental parameters were selected according to the literature [12,13,14] for thermoset systems.

### 2.3. Determination of the Thermal Conductivity

The thermal conductivity of non-cured and cured liquid silicone rubber was determined employing two methods suitable for each state of cure. While the individual components A and B and the mixture (A + B) 1:1 were characterised by employing the transient line-source method, crosslinked compression-moulded (160 °C, 10 min) samples were evaluated by employing a steady-state-based guarded heat flow meter. These two methods are distinct in the sense that while transient approaches rely on applying a short energy pulse to the sample and evaluating the transient temperature rise, steady-state techniques determine the thermal conductivity after the sample reaches thermal equilibrium. However, they are comparable, as reported by Kerschbaumer et al. [15] when investigating natural and synthetic industrial rubber compounds.

#### 2.3.1. Transient Line-Source Technique

Determination of the non-cured samples’ thermal conductivity was conducted according to the standard ASTM D5930-09 [16] in a K-System II device (Advanced CAE Technology Inc., Ithaca, NY, USA). This method relies on placing a line source in close contact with the sample and measuring the rate propagation of the heat released by the line source radially through the sample. Since the rate of heat propagation is related to the thermal diffusivity of the sample and the temperature rise of the line source varies with the logarithm of time, the sample’s thermal conductivity can be determined.

In practical terms, LSR samples (in triplicate) were placed inside the device’s cylinder (9.3 mm diameter), and the line source (1.3 mm diameter) was inserted into the cylinder through the LSR sample. The whole cylinder was then heated to the desired temperature, thermally stabilised (15 min), and the measurement proceeded with a predefined amount of energy being dissipated by the immersed line source. Due to the released heat, the temperature increment ΔT as a function of time ($t_{1}$→$t_{2}$) [15] was recorded and the slope *C* (K^−1^) was calculated:(4)$$C=\frac{ln\frac{t_{2}}{t_{1}}}{\Delta T}$$

The thermal conductivity λ (W·m^−1^·K^−1^) was calculated according to the Fourier’s heat transfer equation for a radial system:(5)$$\lambda =\frac{\kappa }{4\pi }\phi C$$
where ϕ (W·m^−1^) is the line source’s heat flow per unit length. The dimensionless calibration factor κ is obtained after calibration conducted against a reference material with known thermal conductivity, as detailed in the standard [16], also covering the finite probe dimensions, contour effects, and other non-linearities [15]. According to the standard ASTM D5930 [16], this method has an accuracy of ±7%.

For each selected temperature, the same procedure was adopted. For the individual components A and B, thermal conductivity was measured at 80 °C, 100 °C, 120 °C, 140 °C, and 160 °C. However, to avoid the curing reaction, the (A + B) 1:1 mixture was tested at 60 °C, 70 °C, and 80 °C.

#### 2.3.2. Guarded Heat Flow Meter Method

This method is based on the heat flux through a sample that is placed between a heat source and a heat sink. The 2 mm compression-moulded samples (triplicate) were mounted in a thermal conductivity tester DTC-300 (TA Instruments, New Castle, DE, USA) and tested after reaching thermal equilibrium at 60 °C, 70 °C, 80 °C, 90 °C, 120 °C, 140 °C, and 160 °C. By measuring the heat flow $\dot{Q}$ (W) through a specimen with thickness *d* (m) and area *A* (m^2^) and the temperature difference ΔT across the sample, the thermal conductivity λ (W·m^−1^·K^−1^) was calculated employing the Fourier’s heat transfer equation for linear systems:(6)$$\lambda =\frac{\dot{Q}\;\cdot \;d}{A\;\cdot \;\Delta T}$$According to the standard ASTM E1530 [17], this method has an accuracy of ±5%.

### 2.4. Determination of the Specific Volume

For the characterisation of the specific volume, one single method was employed, the so-called piston-die or piston-based isobaric method. It has been shown [18] that the piston-based method differs from the confining-fluid approach by a maximum of 4%; thus, the data reported in this work can be safely compared to specific volumes determined elsewhere.

The specific volume dependence on temperature and pressure was characterised in a PVT 100 dilatometer (SWO Polymertechnik GmbH, Krefeld, Germany). This device employs a piston technology and is based on a cylindrical metal cavity where the sample is placed under pressure between two pistons tightly fitting the cylinder. The change in volume Δv(p,T) is then calculated under specific pressure and temperature conditions as(7)$$\Delta v\left(p,T\right)=\frac{\Delta l\;\cdot \;\pi \;\cdot \;r^{2}}{m}$$
where *l* is the piston displacement length, *r* is the cavity’s radius, and *m* is the sample’s mass. For these measurements, friction of the sample with the cavity wall was neglected and the temperature was varied from 50 °C to 200 °C (heating at 2 K·min^−1^) at increasing constant pressures of 5 MPa, 7.5 MPa, 10 MPa, 15 MPa, 20 MPa, 25 MPa, and 30 MPa (isobaric mode). Crosslinking of the mixture occurred during the first heating at the lowest pressure of the cycle. After each isobaric heating, the cavity was cooled to 50 °C under 5 MPa pressure, and finally the pressure increased to the next isobaric value. A diagram showing the experiment conditions is shown in Figure 1. It is important to highlight that the specific volume at 5 MPa was measured twice: once at the first heating, during which the sample crosslinks; and one a second time, after the 30 MPa step, with the sample fully crosslinked. The specific volume data (in duplicate) were used to fit the Tait model.

### 2.5. Determination of the Curing Kinetics

For the curing kinetics investigation of liquid silicone rubber, methodologies based on a preliminary investigation [19] were employed. Here, the rubber process analyser (RPA, rheology-based) and the dynamic scanning calorimetry (DSC, calorimetry-based) approaches were applied.

#### 2.5.1. Dynamic Scanning Calorimetry

The curing kinetics of LSR was studied by employing a dynamic scanning calorimeter (DSC1 STAR System Mettler-Toledo International Inc., Greifensee, Switzerland). Approximately 10–20 mg of the A + B 1:1 mixture (triplicate) were placed into aluminium crucibles aiming for maximum possible contact with the crucible’s bottom. Next, the samples were submitted to the following thermal program, under 50 mL·min^−1^ of nitrogen as purge gas, in random order to avoid any unmeasured and uncontrolled disturbances from the laboratory environment and from the device:Isotherm for 3 min at 50 °C.Heating at 2/5/10 K·min^−1^ until 150 °C.Isotherm for 5 min at 150 °C.Cooling at 20 K·min^−1^ until 50 °C.Isotherm for 3 min at 50 °C.Heating at 2/5/10 K·min^−1^ until 150 °C.

The crucible’s mass was measured before and after the thermal program and any sample that underwent mass loss higher than 0.5% was repeated. As advised by Heinze and Echtermeyer [20], crosslinking enthalpy ($H_{x}$) was calculated (mean value) as follows, where β = 2, 5, or 10 K·min^−1^.(8)$$H_{x}=\frac{1}{\beta }\int_{T_{onset}}^{T_{endset}}\dot{Q}\;dT\;$$

The curing conversion α for this calorimetry approach was calculated as the released heat ratio $\dot{Q}$ according to Equation (Equation 9). The curing speed, or crosslink conversion rate dα/dt, was calculated employing the differentiate function from Origin 9.0G software (OriginLab, Northampton, MA, USA), i.e., the derivative at a given point was computed by taking the average of the slopes between the point and its two closest neighbours.(9)$$\alpha \left(t\right)=\int_{t_{onset}}^{t}\dot{Q}\;dt\;{\left(\int_{t_{onset}}^{t_{endset}}\dot{Q}\;dt\;\right)}^{-1}$$

The activation energy for this calorimetry-based approach was calculated following the integral isoconversional method for non-isothermal experiments [21,22] depicted in Equation (Equation 10).(10)$$\ln\left(\frac{\beta_{i}}{T_{\alpha ,i}^{1.92}}\right)=Const-1.0008\;\cdot \;\left(\frac{E_{\alpha }}{RT_{\alpha }}\right)$$

#### 2.5.2. Non-Isothermal Rotational Rhometry

Rubber process analyser (D-RPA 3000 Montech Werkstoffprüfmaschinen GmbH, Leipzig, Germany) equipment, or a moving die rheometer (MDR), was employed as a rotational oscillatory rheometer to characterise the curing behaviour of LSR under various curing conditions. For this set of experiments, in triplicate, three heating rates (β = 2, 5, and 10 K·min^−1^), and three shear conditions (0.4383, 4.383, and 13.1476 s^−1^) were employed. The shear conditions were realised by changing the shear frequencies (1, 10, and 30 Hz) and keeping the shear strain amplitude (0.5°). The shear amplitude (0.5°) is within the non-linear viscoelastic range, as demonstrated by our group in a previous work [8], and it is expected that G″ is higher than G′. Thus, the gel point was defined [23] as the time or temperature at which G′ = G″. This means that, above this point, the LSR behaves predominantly as an elastic solid (G′ > G″).

The conversion rate was determined as stated in Equation (Equation 11) as follows, where *M* is the transmitted torque at time *t*, ML is the minimum transmitted torque, and ΔM is the difference between the maximum and the minimum transmitted torques:(11)$$\frac{d\alpha }{dt}=\frac{1}{\Delta M}\frac{d(M-ML)}{dt}$$The activation energy was calculated following the integral isoconversional method for non-isothermal experiments depicted in Equation (Equation 10). Fitting of dα/dt followed the method described next.

#### 2.5.3. Curing Kinetics Non-Linear Fitting

Regarding the kinetic model function f(α), it is widely accepted [23,24,25,26] that the crosslinking reaction of silicone rubbers follows the autocatalytic model, first proposed by Šesták-Berggren [27] in the form of Equation (Equation 12) and further modified by Kamal [28] to include the temperature dependence, as seen in Equation (Equation 13):(12)$$f\left(\alpha \right)=\alpha^{m}{\left(1-\alpha \right)}^{n}$$(13)$$\frac{d\alpha }{dt}=\left(k_{1}+k_{2}\;\cdot \;\alpha^{m}\right){\left(1-\alpha \right)}^{n}$$
where *m* and *n* are the reaction orders and $k_{1}$ and $k_{2}$ are the Arrhenius rate constants. For the present study, a third version of the autocatalytic model was assumed, where $k_{1}$ and $k_{2}$ are taken as constants with activation energies $E_{\alpha ,1}$ and $E_{\alpha ,2}$ in a way that the final kinetic model equation can be written as(14)$$\frac{d\alpha }{dt}=\left[A_{1}\;\cdot \;exp\left(\frac{-E_{\alpha ,1}}{R\;\cdot \;T}\right)+A_{2}\;\cdot \;exp\left(\frac{-E_{\alpha ,2}}{R\;\cdot \;T}\right)\alpha^{m}\right]\;\cdot \;{\left(1-\alpha \right)}^{n}$$

The subscript 1 denotes the $n^{th}$-order contribution to the crosslinking model, while the subscript 2 denotes the autocatalytic contribution. Similarly, *n* is the reaction order for the $n^{th}$ model, and *m* is the reaction order for the autocatalytic part.

Fitting of the dα/dt data according to the Kamal model was performed for conversion values from 0.05 to 0.95. Initially, it was assumed [23] that $E_{\alpha ,1}=E_{\alpha ,2}=E_{\alpha }$ as calculated by applying the isoconversional approach. This $E_{\alpha }$ value was then implemented as the activation energy at the preliminary fitting routine and the parameters $A_{1}$, $A_{2}$, *m*, and *n* were determined by employing the Levenberg–Marquardt algorithm [29] or damped least squares method. After the first preliminary fitting, the previously calculated parameters were employed as initial guesses for the final calculations of the kinetic parameters $A_{1}$, $A_{2}$, *m*, *n*, $E_{\alpha ,1}$, and $E_{\alpha ,2}$. All fitting procedures were performed utilising Python coding language (Python 3.8, PyCharm Community Edition 2021.2.2, Jet Brains, Prague, Czechia). The code was built with the aid of the chatbot and virtual assistant ChatGPT (OpenAI, San Francisco, CA, USA) version GPT-3.5.

### 2.6. Comparison Routines—Simulation Setup

In order to compare the several characterisation methods presented in the previous sections, simulations (software SIGMASOFT v6.1.0.0, 64-bit release in 2024) were carried out employing different sub-datasets for viscosity, specific heat capacity, and curing kinetics. In this sense, five different LSR material data arrays were considered, which are detailed in Table 1.

For the viscosity sub-dataset, the viscosity variation with temperature and shear rate was compared between the data obtained via LAOS or HPCR; therefore, datasets A and B were compared. The comparison between measuring $c_{p}$ via the sapphire method or the modulated DSC approach is made between datasets B and C for the first heating (considering the curing step) and B and D (linear relationship of $c_{p}$ with temperature, without crosslinking). Finally, to compare the two studied approaches to determine the curing kinetics, datasets B and E were compared: the first employed the sub-dataset obtained from calorimetry-based experiments, and the second employed the dataset obtained from rheology-based tests. For all datasets (A–E), the thermal conductivity was assumed to be a function of temperature for the 1:1 mixture A:B cured sample. Since viscosity, $c_{P}$, and the curing parameters can vary, when these are set as constants for comparing datasets, the HPCR data were employed for the viscosity information; the first heating of the MDSC technique was chosen to represent the specific heat capacity, and the curing kinetics as determined via DSC was used to set the crosslinking characteristics.

Concerning the part geometry and the simulation settings, the simulation was carried out employing a rubber injection moulding multi-cavity mould configuration, as described by Traintinger [30] in his Ph.D. thesis, coupled with a cold running system. The four cavities are all equal, being 161.3 × 110.6 × 6.3 mm^3^, as shown in Figure 2a. Boundary conditions for heat transfer were defined as follows: the rubber was injected at a constant temperature of 25 °C, while the mould walls were maintained at 180 °C throughout the simulation by heating cartridges positioned above and below the cavities; filling time was set to 10 s, and the curing time to 210 s. Steel was defined as the mould material, with thermal conductivity at 100 °C equal to 46.6 W·m^−1^cK^−1^. Heat transfer coefficients were set as the software-suggested default values: 10 kW·m^−2^K^−1^ for steel–steel contact surfaces, and 0.8 kW·m^−2^K^−1^ for liquid silicone rubber–steel surfaces. Solver parameters were kept at the default settings: the time step was automatically controlled using a maximum-volume-filling increment of 1% per step; a maximum of 50 iterations per time step was allowed; and a convergence tolerance factor of 1.0 was applied. These settings ensured numerically stable and accurate results without unnecessary computational overhead.

A generic injection moulding machine was chosen from the SIGMASOFT database (SIGMA/generics160-50), with 1600 kN maximum clamping force, 1300 bar maximum injection pressure, and a 3000 bar·s^−1^ maximum pressure increase rate. This virtual injection moulding machine is characterised by a 60 cm^3^ nozzle volume and a 50 mm piston diameter, leading to a maximum dosage volume of 510.51 cm^3^. Three heating cycles were defined and the data were gathered at the fourth cycle. It is important to note that no curing degree limit was predefined in the simulation to determine the cycle time. The total curing time was fixed at 210 s to enable tracking of the curing development throughout the entire moulding cycle. References in later sections to curing degrees between 75% and 95% refer only to typical demoulding conditions for LSR [31], where the residual internal heat after ejection promotes the completion of curing.

To compare the various datasets, pressure, temperature, and curing degree information at the sensors (blue squares at the sprue and in the cavity in Figure 2a) were obtained and analysed for each pair. Sensor 1 in the sprue (mould inlet) was placed to check the flow characteristics at the very beginning of injection, being important to derive the necessary parameters for the injection moulding machine. In Figure 2a, sensor 1 is shown above the running system, in the position where the polymer flows from outside the mould into the sprue. Sensor 2 in the part was placed close to the cavity entrance to monitor the flow properties. This sensor is positioned close to the wall opposing the flow entrance into the cavity. This spot in the cavity will face LSR flow during most of the filling stage, being an adequate place to locate the sensor. Only one sensor was employed, since the mould is considered balanced concerning the flow into the four cavities, i.e., the analysis of one cavity is expected to derive the same conclusions as for the other three. In addition, qualitative information concerning the filling pattern was also employed to compare the effect of using different material data to run the simulations.

Meshing was accomplished via setting equidistant mesh parameters, with 0.5 × 0.5 × 1.0 mm^3^ elements, resulting in 937,246 cells composing the cavity. Details of the mesh are shown in Figure 2b. For the gate and the runners, 1.6 × 1.6 × 1.6 mm^3^ elements were set. All simulations were performed using the same refined mesh resolution across every case to ensure comparability of the results. Using an identical fine mesh for all scenarios helps ensure that any observed differences in outcomes arise from the different material property datasets rather than from discretisation artefacts. While a dedicated mesh convergence study for each case was not performed, we emphasise that mesh sensitivity is an important consideration in general. Here, we mitigated that risk by employing a high-quality, uniform mesh for all runs. This consistent meshing strategy provides confidence that the differences reported in fill behaviour and outcomes are attributable to the material property differences under investigation, not to mesh resolution effects.

## 3. Results

This section is divided into two parts. The first part introduces the material behaviour in terms of the investigated properties and highlights the main differences between the applied methods, reasoning them in terms of macromolecular and structural behaviour. Secondly, the material behaviour is translated into material data (sub-datasets) and implemented in injection moulding simulations, whose results are critically compared.

### 3.1. Specific Heat Capacity Trend

The specific heat capacity of an 1:1 A:B mixture of liquid silicone rubber was determined byb employing the modulated temperature dynamic scanning calorimetry approach and compared to the widely applied and standardised sapphire method. The generated heat during the MDSC run is shown in Figure 3, where exothermal events are represented by peaks pointing upwards. As explained before, MDSC is able to decompose the total heat (as measured in conventional DSC devices) into reversible and non-reversible components. For liquid silicone rubber, the crosslinking appears as an exothermal signal below 100 °C, with 262 mJ released energy, in the total and non-reversible heat curves. Since crosslinking is a kinetic phenomenon, it appears as released non-reversible heat. However, no exothermal signal is present in the reversible heat. This is the first indication that the specific heat capacity, i.e., the amount of energy (J) necessary to increase the temperature of 1 g of material by 1 K, does not change during the crosslinking reaction. Stark, McHugh, and co-workers [13,14] observed a different behaviour while studying the curing of an epoxy–amine resin without reinforcing fillers. The authors observed a first increase in $c_{p}$ (consequence of a change in the reversible heat, Figure 4 in [13]) during the curing reaction and before the onset of vitrification, which for the epoxy–amine resin is due to a specific effect of primary and secondary amine reactions. Subsequently, a sharp decrease in $c_{p}$ was observed for these epoxy–amine systems, which can be attributed to the abrupt reduction in macromolecular segmental mobility, directly related to the specific heat capacity [32]. For liquid silicone rubber, on the other hand, the hydrosilylation reaction and the resulting crosslinked network do not seem to considerably influence the reversible heat.

The specific heat capacity of liquid silicone rubber during crosslinking was then determined based on the reversible heat for the MDSC approach and considering the sapphire reference for the standard ASTM E1269. For this comparison, the first heating in the DSC program is related to the uncured sample, during which the crosslinking reaction occurs, whereas the second heating step involves the already crosslinked sample. In Figure 4, the standard sapphire method (blue lines) shows a change in the $c_{p}$ for the first heating, which is related to the way this value is calculated, i.e., based on the total released heat. Since the crosslinking reaction is exothermal, the sample heats up at a faster pace compared to the reference pan of the DSC device, resulting in a decrease in the specific heat capacity. As the curing conversion reaches 100%, the rate of heat release decreases and the $c_{p}$ raises, returning to a stable value after the crosslinking reaction is completed. It is important to notice that the specific heat capacity above 150 °C for the first heating returns to the trend established below 100 °C (before the reaction onset), suggesting that the $c_{p}$ change experienced during curing is an artefact of the calculation method, not a consequence of chemical or microstructural change in the liquid silicone rubber. Indeed, it is not reasonable to suggest that the crosslinking reaction leads to a decrease and posterior increase in $c_{p}$ as a result of any chemical or microstructural change occurring exclusively due to a connection of adjacent poly(siloxane) macromolecules. One must consider, however, that the $c_{p}$ trend as shown by the full blue line reflects the exothermal nature of crosslinking, which is an important aspect to account for in injection moulding simulation. For the completely cured sample, the second heating shows a linear $c_{p}$ that only increases with temperature.

The modulated temperature DSC method results in $c_{p}$ values, as shown in Figure 4 by the red lines (one replicate, the coefficient of variation among the triplicates was 7%). As expected, $c_{p}$ is a linear function of temperature for the first and second heating steps. The linear behaviour of $c_{p}$ results from the linearity of the reversible heat (red line) over temperature in Figure 3. The sample during the first heating does not experience a change in $c_{p}$ throughout the whole temperature range, showing that the crosslinking reaction alone does not change the specific heat capacity, as shown before for the sapphire method. In the case of liquid silicone rubber samples, there are two main reasons for why the curing reaction plays an almost insignificant role in changing the $c_{p}$. The first and major reason is the presence of a high filler volume (silica), which significantly controls segmental mobility due to interaction with poly(siloxane) macromolecules and energy absorption. Since both uncured and cured samples have the same filler content (they are the same sample, but with distinct physico-chemical states), no change in $c_{p}$ is observed. The second reason why the curing reaction does not lead to a change in $c_{p}$ is the macromolecular arrangement prior and after crosslinking. Before curing, poly(siloxane)s are already highly entangled, since the molecular weight surpassed the critical molecular weight for entanglements. After curing, entanglements remain, and crosslinking bridges build between macromolecules via poly(siloxane) oligomers. Even though these bridges substantially increase the molecular weight, turning the network that was only crosslinked before into a 3D network of infinite molecular weight, local segmental mobility is still preserved. In another direction, Vera-Graziano et al. [33] observed a decrease in $c_{p}$ after crosslinking poly(dimythylsiloxane). The authors also studied the specific heat capacity of PDMS with similar molecular weight distribution as the one employed in the current study ($M_{n}$ = 66,030 g·mol^−1^, $M_{w}$ = 83,391 g·mol^−1^), but crosslinked via hydrogen abstraction with benzoyl peroxide, not via hydrosilylation. Crosslinking via hydrogen abstraction connects adjacent macromolecules via a short covalent bond, hindering segmental motion and, therefore, modifying $c_{p}$. In this sense, one can understand that as crosslinking density increases, more heat is required to reach the same segmental mobility, leading to a higher $c_{p}$, as observed by the authors [33].

Concerning the absolute values of $c_{p}$, both methods reached the values reported in the literature [34,35,36] for poly(siloxane)s. For both methods, $c_{p}$ for the completely cured sample (second heating) appears as higher, but this is actually an artefact from the variable power asymmetry of the sample and the internal DSC reference. In addition, the methods resulted in apparently distinct $c_{p}$ values (MDSC values are higher than the sapphire one). This baseline shift can be associated with the sample contact with the DSC pan, which is different for every measurement since liquid silicone rubber cannot perfectly sit in the bottom of the pan and the contact is highly dependent on how the operator places the sample inside it. From the linear fittings shown in Figure 4, one can interestingly realise that the variation in $c_{p}$ with temperature (slope) is the same regardless of the method: around 0.0016 J·g^−1^·K^−1^/°C. This is very close to the slope reported by Bicerano [37] when proposing an equation for $C_{p}^{l}$(T) (capital *C* denotes the molar specific heat capacity) for liquid polymers (T > T_*g*_ for rubbery and molten polymers) based on experimental data:(15)$$C_{p}^{l}\left(T\right)\approx C_{p}^{l}\left(298K\right)\;\cdot \;\left(0.613+0.0013T\right)$$

For poly(dimethylsiloxane), Bicerano [37] reported $C_{p}^{l}$(298K) = 117.8 J·mol^−1^·K^−1^, which, considering PDMS polymer’s molar mass (74.01 g·mol^−1^), leads to $c_{p}^{l}$(298K) = 1.59 J·g^−1^·K^−1^. Since this value was obtained for pure PDMS, the effect of incorporated silica in the LSR under study has to be taken into account when comparisons are made. In this sense, since fumed silica has lower $c_{p}$ (around 0.9 J·g^−1^·K^−1^) than PDMS and additionally the presence of silica in a poly(siloxane) matrix causes macromolecule immobilisation, one can expect that the silica-filled PDMS has a lower $c_{p}$ than the pure polymer. Thus, one can conclude that the values reported in Figure 4 are comparable to the one found and calculated by Bicerano [37]. For reference, the specific heat capacity of copper is around 0.3 J·g^−1^·K^−1^.

For LSR, the slope of $c_{p}$(T) before and after the crosslinking event is the same. This indicates that both states probably present similar, if not the same, segment mobility. Indeed, our group studied [38] the variation in $T_{g}$ and free volume for a poly(dimethylsiloxane) rubber after curing, and it was found that curing does not affect the free volume or, thus, the $T_{g}$. The same is very likely occurring here: $c_{p}$ is only a function of temperature, not of the curing conversion. Another hypothesis to explain such behaviour is the fact that this LSR grade is highly filled. In this scenario, the filler properties strongly influence the compound’s thermal properties, regardless of the curing state. Considering liquid silicone rubber, the standard sapphire method and the MDSC approach resulted in comparable values for $c_{p}$ when the intrinsic differences among the measurements were taken into account. Even though the MDSC method represents what is truly happening in terms of segment motion within a certain temperature range, without kinetic artefacts, the sapphire method incorporates information concerning heat release/absorption as $c_{p}$ change, which would have to be additionally loaded for the MDSC approach in terms of enthalpy and the rate of heat release, which can be assumed as the relationship presented in Equation (Equation 16) [39], where α is the curing conversion.(16)$$\dot{Q}=Q_{total}\left(\frac{d\alpha }{dt}\right)$$

This difference concerning the kind of information that is given by each method to the simulation software, i.e., variable $c_{p}$ values including the exothermal reaction or linear $c_{p}$ values and a separate enthalpy of curing, will be further discussed in Section 3.5.2.

### 3.2. Thermal Conductivity Response

Thermal conductivity was determined for the part A, part B, 1:1 A:B mixture in the non-cured state, and the 1:1 A:B mixture in the cured state, aiming to check possible differences between these components. Values (average) for λ are shown in Figure 5 for all components. It is valuable to remember, at this point, that the samples’ 2070 component A, 2070 component B, and the 1:1 mixture A:B uncured were tested by employing the transient line-source method, since it allows the thermal conductivity determination of liquid-like samples. Sample 1:1 mixture A:B cured, on the other hand, had its thermal conductivity measured via a guarded heat flow meter device under steady conditions for the temperature. The measurement temperature range for the 1:1 mixture A:B uncured was limited to 80 °C to avoid the curing reaction. For all samples, the thermal conductivity did not considerably vary with temperature, being statistically the same for the whole tested temperature range and varying around the value λ = 0.2 W·m^−1^·K^−1^, which is typical [40,41,42,43] for LSRs.

The variation in polymers’ thermal conductivity with temperature basically depends on the polymer morphology and its glass transition and melting/crystallisation temperatures. Van Krevelen [44] proposed that the thermal conductivity of amorphous polymers at T > T_*g*_ can be estimated based on the thermal conductivity at the glass transition temperature:(17)$$\lambda \left(T\right)=\lambda \left(T_{g}\right)\;\cdot \;\left(1.2-0.2\;\cdot \;\frac{T}{T_{g}}\right)$$This equation shows that the variation in thermal conductivity above the glass transition is low, slowly decreasing with the temperature increase. However, van Krevelen proposed this equation considering amorphous polymers in general, being roughly an estimate about the thermal conductivity. Zhong and co-workers [45] compared van Krevelen’s assumption with λ data for poly(ethylene) and poly(propylene) and observed a flatter change in thermal conductivity with temperature. For cis-1,4-poly(isoprene) (natural rubber, NR), the relationship proposed in Equation (Equation 17) is reasonable, as demonstrated by Eiermann and Hellwege [46]. Above 50 °C, however, the authors reported a plateau for the thermal conductivity. Kerschbaumer and co-workers [15] also reported a plateau for λ(T) when studying rubber compounds based on cis-1,4-poly(isoprene) (NR), poly(styrene-co-butadiene) (SBR), poly(acrylonitrile-co-butadiene) (NBR), hydrogenated NBR, and ethylene propylene diene monomer rubber (EPDM), all of them highly filled with either carbon black, white fillers, or a combination of both. Based on a molecular dynamics simulation, Xu and co-workers [47] determined λ(T) for a poly(siloxane) (relative molecular mass = 28,000 g·mol^−1^) and also reported a plateau between −73 °C and 226 °C. The authors argue that the thermal conductivity is the same because the phonon density of states does not change with temperature (see Figure 8 from [47]), meaning that the temperature does not affect the states available for the phonons to occupy. Based on the reported findings, the results presented in Figure 5 for an independence of λ with temperature are reasonable. Poly(siloxane) macromolecules at the studied temperature range are far above their glass transition and melting temperatures and do not experience any thermal transition or chemical change. Thus, it is fair to assume that, as previously reported, neither the phonon density of states (dependent on the backbone atoms, silicon and oxygen) nor the thermal conductivity change. Furthermore, since the liquid silicone rubber grades are highly filled with silica, it is also fair to consider that the filler controls phonon transmission in these samples, in accordance with what has been reported in the aforementioned literature.

Between the individual components A and B, no significant difference was detected, indicating that these two parts have similar thermal conductivities. The fact that part B has a higher deviation may be connected to the presence of the crosslinker, which is a poly(siloxane) oligomer with lower molecular weight when compared to the main poly(siloxane) base polymer. The uncured mixture shows a tendency for presenting a similar λ when compared to its single constituents. The cured sample, on the contrary, presented a lower thermal conductivity than the uncured mixture and an inclination to lower λ than the individual parts. This finding is contrary to what was observed by Cheheb and co-workers [48], who identified a 10% increase in the thermal conductivity for crosslinked rubber compounds when compared to the uncured counterpart. For the present scenario, considering that the samples’ thermal conductivities are dominated by the filler thermal properties and that curing of LSR connects two adjacent macromolecules via a linkage that is oligomeric (the size of the Si–H-based crosslinker), a decrease in λ does not seem reasonable. Thus, this apparent decrease in thermal conductivity after crosslinking may be associated with the different measurement method employed between the cured and the uncured samples. In order to clarify this aspect, the averages ($\bar{\lambda }$) for the whole temperature range are shown in Table 2, where the standard deviation (σ(λ)) and the error associated with each measurement according to Kerschbaumer and co-workers are also stated [15]. Furthermore, since the transient line-source method requires the filling of a cylinder with liquid sample to perform an analysis of λ, any air bubbles would cause an increase in the sample’s thermal conductivity, probably leading to the higher values observed for the uncured samples.

For simulation, the values reported in Figure 5 have practical consequences in terms of how the thermal conductivity data can be input into the routine. The first outcome is that the individual components (either A or B) can be analysed in order to gather thermal conductivity data for the simulation, with the advantage to cover a larger temperature range without crosslinking. Following this outcome, a completely cured sample could also be employed to measure the thermal conductivity of an LSR grade to be further injection-moulded, however, demanding a previous sample preparation. Lastly, if required, either by the simulation software or to save computational time, the thermal conductivity can be reliably considered as constant within typical liquid silicone rubber injection moulding temperatures.

### 3.3. Specific Volume or pvT Behaviour

The variation in specific volume with temperature and pressure is peculiarly important for liquid silicone rubber due to the fact that it experiences considerable thermal expansion during processing [31]. For the studied LSR grade, the specific volume under various isobaric conditions as a function of the temperature is presented in Figure 6, consistent with the scarce values reported in the literature [5]. As was expected, for all pressures, the specific volume increases with temperature. This expansion is driven by the vibrational motion of poly(siloxane) macromolecules, which gain more energy as the temperature increases and, therefore, occupy more volume due to increased vibration. Under isothermal conditions, though, the effect of pressure is on decreasing the specific volume and, in this case, it is the packing volume (empty space between the macromolecules) that is reduced. During injection moulding, liquid silicone rubber undergoes slight compression during injection (A-B segment in Figure 6) and heating mostly due to shear-related dissipation. Injection is over at point B, when the pressure is equalised and the cavity is completely filled. Heat transfer from the hot mould to the uncured LSR occurs, leading to heating and thermal expansion in steps B-C. Due to the fact that filling is normally carried out subvolumetrically, LSR is further heated under isochoric pressure to point D. The cavity conditions are held until the part is fully cured, being ejected to point E and further cooled down back to point A.

By combining the effect of pressure and temperature, one can realise that thermal expansion is hindered by the pressure increase, as the slope of the plotted lines in Figure 6 get flatter as pressure rises. The slopes were determined via linear fitting and plotted over the applied pressure in Figure 7. Volume change decreases almost linearly with the pressure, showing that vibrational mobility triggered by a temperature increase is hindered by the reduced empty space between macromolecules.

From the data presented in Figure 6, the Tait model (Equation (Equation 18)) coefficients were determined based on a least squares method and are shown below. The coefficients $b_{1m}$ (m^3^·kg^−1^) and $b_{2m}$ (m^3^·kg^−1^) represent the dependence of the specific volume at zero pressure on pressure and temperature; $b_{3m}$ (Pa) corresponds to the pressure dependency of parameter B(T), and $b_{4m}$ (K^−1^) adds the temperature correlation [49]. These are model parameters for the molten domain:(18)$$\left\{\begin{array}{c} v\left(p,T\right)=v_{0}\left[1-C\;\cdot \;ln\left(1+\frac{p}{B}\right)\right]+v_{t}\\ v_{0}=b_{1m}+b_{2m}\;\cdot \;\bar{T}\\ B=b_{3m}\;\cdot \;exp\left(-b_{4m}\;\cdot \;\bar{T}\right)\\ v_{t}=0,\;for\;T>T_{g},T_{m} \end{array}\right.$$
where v(p,T) is the specific volume in terms of pressure and temperature, $v_{0}$ is the specific volume at zero pressure, and *C* is a universal constant taken as 0.0894. $v_{t}$ describes the volume change due to crystallisation or glass transition.(19)$$\left\{\begin{array}{c} b_{1m}=6.091\times 10^{-4}\;m^{3}\;\cdot \;{\mathrm{kg}}^{-1}\\ b_{2m}=9.025\times 10^{-7}\;m^{3}\;\cdot \;{\mathrm{kg}}^{-1}\\ b_{3m}=1.278\times 10^{8}\;\mathrm{Pa}\\ b_{4m}=3.367\times 10^{-3}\;K^{-1} \end{array}\right.$$
These parameters will be further utilised as input data for the simulation comparisons carried out next.

### 3.4. Crosslinking Kinetics Characteristics

The curing kinetics of LSR was studied via a calorimetric approach (DSC), as well as a rheology-based approach (rubber process analyser, RPA). In this section, the results obtained from both approaches are first presented separately and then compared in terms of the calculated Kamal model (Equation (Equation 14)) parameters.

#### 3.4.1. Curing Kinetics via DSC

When studied under a calorimetric perspective, liquid silicone rubber mixture (1:1 A:B) samples show a heat flow behaviour as shown in Figure 8 (all replicates are shown in the plots, aiming to also demonstrate the method’s repeatability). For all heating rates, the exothermic peak related to curing is apparent at temperatures higher than 90 °C. As the heating rate increases, the temperature at which the curing speed is maximum (peak apex) shifts to higher values due to the faster heating of the sample (thermal inertia effects); i.e., since the sample is kept under a certain temperature for a longer time, curing occurs at lower temperatures. The crosslinking enthalpy (as defined by Equation (Equation 8)) is the same for all samples and heating rates, being 8.15 ± 0.27 J·g^−1^. This quantity is important for injection moulding simulation, since it represents an internal heat source, which is incorporated into the constitutional equation of energy conservation as $\dot{Q}$.

From the shape of the exothermic peaks, one can realise that the speed of heat release is proportional to the extent of curing or conversion α. Thus, the conversion α was calculated and the conversion speed dα/dt was determined and plotted over temperature and over conversion in Figure 9a and Figure 9b, respectively. From the conversion rate data, it is possible to realise that the maximum speed of curing occurs at around α = 0.6 for all heating rates. This is a good indication that the curing mechanism does not change with the heating rate. In addition, it is typical of autocatalytic curing reactions to reach the maximum reaction rate at intermediate conversion values, while n^*th*^-order-based reactions have the maximum conversion rate at the beginning of the curing reaction [50]. Thus, it is appropriate to fit the conversion rate data to the autocatalytic, or Kamal, model.

When the kinetic model of a certain chemical reaction is not known, it is usual to apply isoconversional approaches to determine kinetic parameters, such as the activation energy, which are model-independent. The variation in activation energy with the curing extent is shown in Figure 10b, while Figure 10a shows an example of a plot utilised to calculate E_*a*_ for each conversion step. As for the case of high-consistency silicone [19], the activation energy is fairly constant and has an averaged value of 134.5 kJ·mol^−1^. This value is consistent with the one found by Hernández-Ortiz and Osswald [51], who studied an injection moulding grade of LSR; and it is slightly greater than the values (between 80 kJ·mol^−1^ and 100 kJ·mol^−1^) obtained by Ke et al. [52], who studied a liquid silicone rubber with lower molecular weight than the one investigated in the present work. The activation energy as determined in the isoconversional approach is the overall energy barrier of the reaction under study. This means that it shows the thermodynamic boundary of curing, regardless of the number of steps involved in the curing mechanism or the type of mechanism. Next, two values of E_*a*_ will be presented, one for the n^*th*^-order contribution and one for the autocatalytic part of the kinetic model, not being related to mechanistic steps of the crosslinking reaction. Thus, for the sake of this study, it is assumed that the curing reaction occurs in one single step with one set of kinetic parameters, neglecting any effects related to the diffusion of reactive species that would ultimately lead to more than one activation energy determination [26].

With the data presented until this point, it is possible to fit the conversion rate as shown in Figure 9b to the Kamal model to obtain the kinetic parameters A_1_, A_2_, E_1_, E_2_, m, and n. To facilitate the comparison with the rheology-based approach, this is first introduced next, and the fitting comparison is made further in this section.

#### 3.4.2. Curing Kinetics via RPA

While for the calorimetry-based approach, the released heat was employed as a measurable quantity to quantify the curing extent, for the rheology-based method, a measurable mechanical response is used to characterise the crosslinking development. Thus, the transmitted torque was recorded for the three heating rates and under three shear rates, and plotted over temperature, leading to the graphics shown in Figure 11. For all heating rates and shear rates, the torque increases sharply as the curing reaction starts due to the increase in molecular weight that imposes a high restriction on the set oscillation. As the heating rate increases, the onset temperature of curing is shifted to higher values, as occurred in the calorimetric approach. The effect of shear rate is apparent in the torque values: the higher the imposed shear rate (higher frequency of oscillation), the lower the transmitted torque. This is due to shear thinning effects, as thoroughly detailed in a previous publication of our group [8]. There is no significant effect of the shear rate in the onset of curing; i.e., neither an activation of the curing reaction nor an acceleration (higher curing speed, as shown next) by the increasing shear rate, as reported by Ziebell and Bhogesra, was observed [53].

One important feature of the torque increase due to curing is the steady and slow increase after the maximum curing rate, which in some cases lasts until the end of the measurement (200 °C). This steady increase in the torque during RPA analyses is known as the marching modulus, which is a phenomenon related to the filler–filler interactions between silica particles, but is still not fully described in the literature [54]. The occurrence of a marching modulus is detrimental to the characterisation of curing kinetics, since a definitive end (α = 1) of the chemical reaction has to be established. In this sense, the end of curing for these measurements was set at the temperature of maximum transmitted torque.

The speed of a reaction, or the conversion rate, is shown in Figure 12 as a function of temperature (a–c) and of the conversion (d). As stated before, the increase in shear rate does not impose a significant change neither in the onset of curing nor in the curing speed. Indeed, it is plausible to argue that stronger shear heating is likely to occur when the shear rate is increased, leading to a faster curing reaction. However, probably due to the device robustness, this difference could not be monitored. It is interesting to note that the maximum curing speed occurs at low conversion values, which reflects the steep increase in torque right after the onset of curing. This already indicates distinct reaction orders *m* and *n* when compared to the calorimetric approach, since there is a close relationship [50] between the conversion at which the curing speed is at maximum and the reaction orders(20)$$\alpha \left(max\left(\frac{d\alpha }{dt}\right)\right)=\frac{m}{m+n}$$Another important observation is the variability in dα/dt for the same measurement settings, as represented by the shaded areas around the average (solid lines in Figure 12d). By comparing with the DSC experiments, the RPA method is more prone to variation within replicates, since the sample mass is higher and the measurable quantity is a mechanical property, whose sensibility is lower than for the modern calorimeters.

Following the isoconversional approach, the activation energy for the curing as measured by the rheology-based approach was calculated and is expressed in Figure 13 for all three shear rates. Again, the increase in shear did not impose any change in the activation energy, which showed a slight decrease with the conversion. One can easily realise once more that the reproducibility of DSC measurements was higher than the RPA ones by the variation in the activation energy shown in Figure 13. In addition, E_*a*_ as determined by DSC, is more constant when compared to the one characterised via RPA, suggesting again that the kinetic parameters of the Kamal model will likely be different. Statistically, the activation energies determined from the different approaches are distinct, as also reported by Bardelli et al. [23]. However, Bardelli and co-workers reported higher values for the calorimetric analysis when compared to the rheological one. This difference might be related to the different LSR grade studied by the authors, which was the low-viscosity Sylgard184. This LSR is also cured via hydrosilylation reaction, but it is employed as a dielectric gel that is applicable for sealing and protecting various electronic devices.

The time and temperature at which the viscous (G″) and the elastic (G′) contributions are equal are normally employed to determine the gel point for systems that undergo a liquid–gel transition [55]. For the samples under study, these values are reported in Figure 14. The shear rate also has no effect on the gel point, as evidenced by the lack of a trend in terms of gel time and gel temperature. In addition, it is more interesting to notice that the elastic modulus surpasses the viscous one in the very beginning of the curing detection in the RPA: by comparing with Figure 11a, at 2 K·min^−1^, the gel temperature is around 100 °C, which is at the very onset of curing. This behaviour was noticed in the literature by Harkous et al. [55] and more recently by Weißer et al. [56], who argued for the existence of a physical gel (derived from the filler–filler interactions and macromolecular entanglements) before a chemical gel is formed due to curing. This argument is confirmed by the present study, since this physical gel was proven to exist in a previous publication of our group [8], where the rheology of LSR was thoroughly investigated: the physical gel is disturbed under high shear amplitude and G″ is higher than G′ at 10% deformation.

Thus, one can conclude that the classical definition of a gel point very likely cannot be applied to highly elastic and filled systems as injection moulding grades of LSR. Instead, the standard definition of optimum cure time, widely employed in the rubber industry, may seem to be more appropriate here. In this case, the optimum cure time is determined as the time at which 90% of the maximum torque during RPA measurements is reached [19]. Here, it is convenient to point out that the calorimetry-based method does not offer the possibility to determine the gel point or the optimum cure time. However, as demonstrated before and compared to RPA next, it is able to characterise the curing kinetics of LSR in a sense that the conversion can be determined during simulation to predict the optimum cycle time [57].

#### 3.4.3. Fitting and Comparison Between DSC and RPA

In order to compare the methods available to characterise the curing kinetics of LSR for injection moulding simulation, qualitative and quantitative analyses are discussed next. Qualitatively, the conversion over temperature is compared in Figure 15 for the samples tested at 2 K·min^−1^. Only the samples at the lowest shear rate condition are shown, since these are closer to static conditions, as experienced during DSC runs, and because no significant qualitative difference was observed for other shear conditions. From this plot, it is possible to clearly see that the DSC samples show higher values of conversion at lower temperatures, i.e., the reaction is detected first. This observation is explained by the fact that, while the calorimetric approach is able to detect the heat released since the very beginning of the curing reaction, the rheology-based method requires that a minimum (according to the device sensitivity) torque increase occurs to then be detected by the device. This means that the reaction response of one mol of Si–H-based crosslinker with one mol of vinyl-modified chain ends of poly(siloxane) oligomers is more strongly detected by its released heat, not by the increase in molecular weight that leads to shear resistance (increase in the transmitted torque). While sensitivity is the key issue here, the fact that enough macromolecules have to be connected to enhance shear resistance also plays a role in the detected differences. In this sense, one can argue that the reaction happens in the RPA even before a torque increase is detected, but the molecular weight did not increase enough to impose resistance higher than the device’s transducer sensitivity. When the torque resistance is enough to reach the device sensitivity, the conversion values are then higher than zero and increase rapidly. Another explanation for such delay is the sample size. DSC measurements require milligrams of a sample, while RPA functions with grams of LSR. Thus, the heat is transferred more easily in the DSC when compared to RPA, affecting the measurement.

In addition, one can notice that the transmitted torque increase is detected at 100 °C, when the conversion according to the DSC approach already reaches almost 50%. This indicates that the rheology-based method is not able to detect the entire curing behaviour, failing to cover the onset and beginning of crosslinking, as also reported in the literature [55]. Furthermore, this approach is likely to have distinct kinetic parameters when compared to the calorimetric method due to the fact that it detects only one part (roughly the second half) of the curing process. To quantify this effect, the heat released by the DSC sample (2 K·min^−1^) up to 100 °C (the temperature at which torque becomes detectable in the RPA at 0.4383 s^−1^) was calculated. This value corresponds to approximately 53% of the total curing heat, confirming that more than half of the reaction progresses before the RPA measurement registers any increase in torque. It is also worth noting that this temperature coincides with the apex of the DSC heat flow curve, indicating that the maximum reaction rate occurs at the beginning of the RPA signal. This behaviour supports the interpretation that the RPA primarily captures the diffusion-controlled or $n^{\mathrm{th}}$-order portion of the reaction. As discussed next, this detection limitation causes the RPA data to fit the Kamal model with an artificially high *n* parameter, whereas the full-profile DSC dataset yields a dominant *m* parameter, as expected for an autocatalytic reaction. In terms of simulation implications, this discrepancy contributes to the observed differences in cycle time prediction, with DSC-based kinetics producing shorter and more realistic cycle times due to earlier onset and higher overall reactivity.

Another important difference between the calorimetric and the rheological approaches, as shown in Figure 15, is the conversion value at which the conversion rate is at maximum. For the calorimetric approach, the maximum occurs at α > 0.5, while for the rheometry-based method, dα/dt reaches its apex in early conversion values. As mentioned before, it is typical for autocatalytic reactions to present the maximum conversion rate at moderate conversion values, since at low conversion, there is less concentration of the product that acts as a catalyst, rendering low reaction speeds. As the reaction proceeds and more product is formed, the reaction is boosted by the catalytic effect, and it reaches its maximum speed at moderate conversion values. The type of reaction that presents maximum speed in early conversion values is the one that follows n^*th*^-order kinetics mechanisms or diffusion-controlled reactions. These reactions present a conversion rate apex at the beginning of the reaction due to the reactant species’ high mobility. As the reaction proceeds, either due to a viscosity increase or formation of a network structure, the reactants are less likely to collide, resulting in a decrease in the curing speed. For the hydrosilylation reaction, the autocatalytic behaviour is explained by the formation of an active form of the catalyst (Pt complex with alkenyl functional group) at the beginning of the curing process that will further catalyse the insertion of the Si–H component [58]. Therefore, first, a substantive amount of the active catalyst has to be formed before the curing speed increases substantially. This is exactly what can be observed in Figure 15 for the blue lines (calorimetric data). On the other hand, the rheology-based approach is only able to detect the curing step after the formation of enough active catalyst, leading to a sudden sharp increase in the conversion, suggesting an n^*th*^-order reaction. This is an important differentiation between the two methods that relies only in the detection of a signal that derives from the curing reaction. The hypothesis that the calorimetric method is able to truly characterise an autocatalytic reaction while the rheological one can only detect the diffusion-controlled portion of the same reaction will be further discussed when the kinetic parameters are presented.

In order to quantitatively compare the approaches, the datasets from Figure 9 and Figure 12 were fitted to the Kamal model (Equation (Equation 14)), and the experimental and fitted plots are shown in Figure 16. The kinetic parameters associated with these plots are detailed in Table 3. From the plots, it is possible to see that the calorimetric data is nicely predicted by the Kamal model with the lowest mean residual. The kinetic parameters associated with the DSC experiments are different from the ones obtained by the RPA approach, as was assumed due to qualitative differences in the conversion rate plots. The difference arises from the fact that the methods detect not only different responses in the curing reaction but they also cover the crosslinking process differently due to method sensitivity. This contrast was even observed when employing more sensitive rheometers, such as the strain-controlled rheometers employed by Harkous et al. [55], Bardelli et al. [23], and by Weißer et al. [56].

The kinetic parameters detailed in Table 3 show that, for the calorimetry-based approach, the reaction order related to the autocatalytic contribution *m* is higher than the contribution of the n^*th*^-order *n* portion of the model. This helps to engage in the hypothesis that the DSC is able to detect the whole curing process, from the formation of the active catalyst to the total consumption of the reactants (α→ 1). However, the rheological approach presents a predominance of the n^*th*^-order contribution, since n>m for all shear rates. This is a result of the abrupt increase in conversion right after the curing onset, typical of reactions that follow an n^*th*^-order kinetic model. Since it is unreasonable to think that the same chemical reaction changes its mechanism depending on the method employed to study it, one can conclude that the way the data is detected plays a role into the fitting to a certain kinetic model. Although without clear explanations, Hong and Lee [24], Harkous and co-workers [55], and Bardelli and co-workers [23] also reported n>m when employing a rheological analysis, while m>n was determined for the calorimetric one. All fittings performed with the data obtained via RPA presented *n* = 3 due to the fact that this parameter was allowed to vary during fitting from 0 to 3. Thus, this is an artefact from the fitting process.

### 3.5. Injection Moulding Simulations

Following the strategy outlined before, processing parameters related to the simulation output are compared next in pairs, aiming to contrast two datasets obtained via distinct experimental methods. For each pair, the most significant injection moulding phase will be assessed, for example, the effect of different curing kinetics parameters are studied only during the solidification (curing) phase, not during filling.

#### 3.5.1. Viscosity Datasets A and B

The viscosity sub-datasets obtained in a previous publication [8] are employed here as input for an injection moulding simulation. For this comparison, datasets A and B from Table 1 were utilised and fitted to the Carreau–Yasuda model (Equation (Equation 21)) [59,60] with the William–Landel–Ferry (WLF) equation (Equation (Equation 22)) [61] to describe the temperature dependency, as follows:(21)$$\eta \left(T,\dot{\gamma }\right)=\eta_{\infty }\alpha_{T}+\alpha_{T}\left(\eta_{0}-\eta_{\infty }\right){\left[1+{\left(\lambda \alpha_{T}\dot{\gamma }\right)}^{a}\right]}^{\frac{n-1}{a}}$$(22)$$log\alpha_{T}=\frac{8.86(T_{0}-T_{S})}{101.6+(T_{0}-T_{S})}-\frac{8.86(T-T_{S})}{101.6+(T-T_{S})}$$
where $\eta_{\infty }$ is the infinite shear viscosity, $\alpha_{T}$ is the temperature dependence given by Equation (Equation 22), $\eta_{0}$ is the zero-shear viscosity, λ is a time constant, *a* is a transition parameter, *n* is the model order, $T_{0}$ is the reference temperature, and $T_{S}$ is the standard temperature. The fitting was conducted by the simulation software and the parameters were obtained for dataset A (viscosity data from oscillatory experiments LAOS) and B (high-pressure capillary rheometer), as shown in Table 4.

The values reported in Table 4 are the parameters that minimise the sum of the squares of the residuals concerning Equations (Equation 21) and (Equation 22). In this sense, no physical meaning will be derived from these values.

The effect of distinct viscosity sub-datasets in filling and curing simulations are analysed next, mostly in terms of the filling phase. Figure 17 shows the flow behaviour of LSR at 20% cavity filling for datasets A and B. It is possible to see that the flow profile is very similar for both simulations, which is the typical parabolic flow characteristic of the velocity profile described by fluid mechanics. The flow behaviour similarity continues during the whole filling stage. From our previous study [8], one can realise that at the highest studied temperature (90 °C), the viscosity values for both methods (LAOS and HPCR) are comparable for the whole shear rate range. Thus, it is expected that the flow front behaves similarly.

At the sprue, the pressure was monitored via a virtual sensor and plotted over the filling time in Figure 18a. Since LSR flows into the mould at 25 °C, higher viscosity for the LAOS dataset is expected due to the fact that at 50 °C, this LSR grade shows $\eta_{LAOS}$ > $\eta_{HPCR}$ [8]. This difference in viscosity, i.e., resistance to flow, leads to a difference in pressure as presented in Figure 18a: a higher pressure is expected for the more viscous sample (dataset B, HPCR). The significant increase in pressure at the sprue indicates that, when designing the injection moulding process, distinct injection force/pressure and speed would have to be defined. In order to further clarify this difference, peak pressures during filling and the integral of pressure over time are displayed in Table 5.

The peak pressure values represent the maximum pressure recorded by the virtual sensor at the sprue during the filling phase, while the integral quantifies the total pressure impulse applied over the same period. Both metrics clearly indicate that dataset B (HPCR) predicts significantly higher flow resistance, requiring increased injection pressure compared to dataset A (LAOS). At this point, it is important to highlight that the viscosity measurements were performed at temperatures below the injection moulding temperature to avoid premature crosslinking during rheological testing. However, the fitted viscosity models (parameters shown in Table 4) were extrapolated to cover the actual process conditions, ensuring compatibility with the simulation environment while preserving the integrity of the experimental data.

In order to check which pressure curve better predicts the actual pressure at the sprue during a real LSR injection moulding process, further experiments would be necessary. Nevertheless, this piece of evidence is the most important one concerning the effect of distinct viscosity sub-datasets, since it shows that processing conditions would also be differently set for each dataset. The pressure profile at the surface of the sprue, running system, and cavities is also shown qualitatively in Figure 19, where the higher pressure for the simulation ran employing dataset B becomes evident. Inside the cavity, there is no significant pressure difference at the end of filling for the simulations.

The comparison between datasets A and B kept the specific heat capacity and thermal conductivity sub-datasets constant, leading to the simulated temperature curve at Figure 18b. During the whole cycle, the temperature measured at the sensor was the same for both datasets, indicating that the viscosity data did not affect how LSR heats up inside the cavity. Indeed, this difference will be observed in the next comparison concerning $c_{P}$.

#### 3.5.2. Specific Heat Capacity Datasets Comparison

The specific heat capacity sub-datasets previously obtained are employed as input for the injection moulding simulations. For this comparison, two pairs of datasets were used: first, datasets B and C from Table 1 to compare the effect of the curing signal in $c_{p}$ data; and second, datasets B and D to compare the magnitude of $c_{p}$ as varying linearly with temperature.

Differently from the comparison concerning viscosity sub-datasets, the specific heat capacity mostly affects the temperature profile and, therefore, the solidification (curing) phase of the injection moulding cycle. In Figure 20, it is possible to check that the pressure at sensor 1, located at the sprue, is very similar for both datasets. The pressure is slightly higher for dataset B (MDSC first heating) because the higher $c_{P}$ values for this sub-dataset (for reference, see Figure 4) cause lower heat flow for the same mould temperature, leading to a higher viscosity. Overall, however, the different specific heat capacity values did not significantly affect the pressure at the sprue. This means that, for a real process, no critical difference would result if the processing set was based on simulation B or C.

The effect of different $c_{P}$ values on the cavity temperature is quantitatively shown in Figure 20b and qualitatively shown in Figure 21a. Dataset C causes higher temperatures throughout the whole cycle than dataset B, which is a direct consequence of the lower $c_{P}$ determined by the sapphire method during the first heating. In addition, it is possible to realise that the input concerning enthalpy (8.15 kJ.kg^−1^) in dataset B was not enough to equalise the temperature to dataset C. This probably occurs due to the fact that the software apparently does not connect the released heat with the curing rate, as stated in Equation (Equation 16), in order to account for the internal heat source. If this is the case, then it is more reliable to input the specific heat capacity dataset determined by the sapphire’s first heating method, since it accounts for the heat release that ultimately raises the polymer temperature.

The curing degree or conversion calculated at the sensor 2 position is shown for the whole cycle duration in Figure 20c, while it is qualitatively shown for the cavity surface at Figure 21b. The curing degree behaviour is directly connected to the temperature profile analysed before; i.e., the higher the temperature at a given cycle time *t*, the higher the conversion, as can be seen in Figure 21c. Both datasets contain the same curing kinetics information but heat up at different paces, leading to distinct curing times. At the end of the filling stage, it is possible to realise from Figure 21 that the simulation carried out with dataset C indicates a curing degree at the cavity surface around 30% in some regions, while the whole cavity’s upper surface shows a negligible curing degree. In this sense, there is no possibility of scorch for any of the scenarios.

The curing degree development during the curing phase is shown in Figure 21 for t = 25 s and t = 55 s (cycle time). At 25 s (Figure 21c), the crosslinking degree difference can be observed, being low close to the cavity entrance for both datasets (lower residence time when compared to the other cavity areas), but still higher for dataset C. At 55 s, the simulation for dataset C showed complete curing for the part, while dataset B’s simulation presented an uncured core. Even though dataset B is not completely cured after 55 s, it is possible to state that the part could still be ejected at this condition, since ejection typically occurs between 75% and 95% completion of cure [31].

In order to compare the specific heat capacity datasets that have a linear relationship of $c_{P}$ with temperature (MDSC first and sapphire second), i.e., without the crosslinking contribution, datasets B and D were compared in terms of their simulation outputs. From the pressure and temperature signals presented in Figure 22, one can realise that both datasets present similar results, with dataset D having slightly higher temperature development than dataset B. The higher temperature related to dataset D reflects in an earlier curing onset, as presented in Figure 22c. The curing degree behaviour for dataset C is shown for comparison, where it is possible to observe not only the effect of the lower average $c_{P}$ but also the heat release contribution (realised as a downward peak at $c_{P}$).

When comparing the temperature at the end of the filling stage, datasets B and D present similar behaviour, as shown in Figure 23. The temperature difference at the center of the part is 1.7 K, which is also the average for the temperature close to the cavity’s upper surface. These datasets will eventually lead to higher temperature differences as the cycle proceeds (Figure 22). In this sense, there is no significant impact of the specific heat capacity in the filling phase, resulting in no scorch or problems to fill the whole cavity within the set filling time.

#### 3.5.3. Curing Kinetics Datasets B and E

To compare the approaches employed to characterise LSR curing kinetics, the Kamal model parameters from Table 3 were employed for the calorimetry and for the rheological approaches. In this sense, the parameters obtained from the lowest shear rate (0.4383 s^−1^) were employed. For clarity, the Kamal model parameters obtained in Section 3.4 are repeated in Table 6 for the calorimetry-based (dataset D) and the rheological approaches (dataset E).

The impact of these different curing kinetics parameters in the filling and curing simulation of LSR injection moulding was studied, and the pressure, temperature, and curing degree development within the injection moulding cycle is shown in Figure 24. As was already demonstrated for the $c_{P}$ study, the pressure measured by sensor 1 did not present significant difference when comparing datasets B (curing parameters based on DSC experiments) and E (curing kinetics characterised via rubber process analyser). This is due to the fact that the position where sensor 1 is located does not reach the curing onset temperature, thus not being affected by the crosslinking kinetics. Therefore, both datasets present the same pressure profile (Figure 24a) at the sprue, leading to possibly the same setups when concerning the injection moulding machine capacity (force, injection speed, etc.).

At sensor 2, which is located close to the cavity entrance, the temperature development (Figure 24b) is also the same for both datasets, reflecting the equal specific heat capacities that were set for these samples. However, since the Kamal model parameters are different, distinct curing degrees were measured at sensor 2, which are shown in Figure 24c. As it was explained before, no scorch occurs during filling, since the curing degree is negligible before t = 10 s at the location of sensor 2. This is shown in qualitative terms by Figure 25: even though the curing onset is already reached at the end of filling (Figure 25a), no significant curing occurs (Figure 25b).

The curing degree inside the cavity follows the conversion trend presented in Section 3.4, where the calorimetric data displayed an earlier onset compared to the rheological one. In addition, the curing degree calculated for dataset B reached 90% 19 s before dataset E, demonstrating once again the problem related to the rheological data concerning the marching modulus (conversion values were calculated based on the torque data, which were sensitive to the marching modulus). However, when one considers that ejection typically occurs between 75% and 95% completion of cure for LSR [31] due to the fact that the remaining internal heat after ejection is capable of completing the part’s curing, datasets B and E experience a 5 s difference in the curing time, reaching a 75% curing degree at around 32 s and 37 s, respectively.

Qualitatively, Figure 26 shows the curing degree development for various cycle times, highlighting the differences in terms of crosslinking onset and speed for the distinct datasets. Between 30 s (Figure 26b) and 40 s (Figure 26c), the part’s upper surface already reaches at least 70% of curing, while the part’s core (blue area evidenced by the cross-section) is still mostly uncured for both datasets.

## 4. Conclusions

This study demonstrated that the thermo-physical and kinetic properties essential for reliable injection moulding simulations of liquid silicone rubber (LSR) can be characterised by different experimental approaches. The specific heat capacity was shown to depend only on temperature, with calorimetric and sapphire-based methods offering complementary perspectives: the former yielding the intrinsic property and the latter embedding enthalpic effects associated with curing. Thermal conductivity was found to remain nearly constant across the investigated temperature range, simplifying its implementation in simulations, while the specific volume was accurately described by the Tait equation, capturing the thermal expansion behaviour critical to mould filling.

In terms of curing behaviour, the classical definition of gel time was shown to be unsuitable for LSR, as early-stage physical gels do not correspond to flow cessation. Furthermore, calorimetric and rheological methods for kinetic characterisation delivered distinct parameter sets when applied to the Kamal model, owing to their different sensitivities. While calorimetry followed the full curing process and supported an autocatalytic description, rheology was limited to the later stages of curing and suggested an alternative $n^{th}$-order representation. Both methods provided acceptable fits to conversion rate data, but calorimetry yielded more consistent results, whereas rheology remains more common in industrial practice.

When implemented in injection moulding simulations, these datasets led to meaningful differences in predictive outcomes. Viscosity inputs primarily influenced injection pressure requirements, though not the overall thermal or early curing profiles. Variations in specific heat capacity impacted thermal predictions and curing rates, with sapphire-based data providing a more realistic representation of heat release during curing. The most pronounced discrepancies arose from curing kinetics datasets, where calorimetric data predicted an earlier onset and faster conversion compared to rheological data, resulting in distinct curing degree distributions and cycle time estimations. These differences strongly indicate the necessity of careful dataset selection, as curing-related properties exert the strongest influence on predictive fidelity. It is important to acknowledge that validation experiments would help establish the relevance of these simulation deviations; however, real injection moulding tests could not be performed in this study due to the unavailability of the proper injection moulding machine and dosing unit for LSR.

Overall, the findings emphasise that while different characterisation techniques can generate valid datasets, their implementation into simulations produces varying levels of accuracy and predictive reliability. For robust process design and optimisation in LSR injection moulding, particular attention must be given to curing kinetics and thermal properties, as they decisively shape curing predictions and cycle time determination.

## Figures and Tables

**Figure 1 polymers-17-03086-f001:**
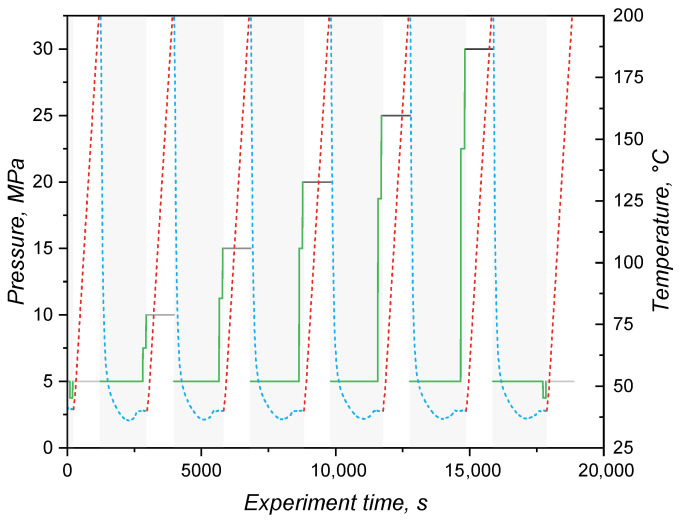
Experimental conditions for the determination of the specific volume under different pressures and for a range of temperatures. The full line represents the pressure imposed on the sample and indicates the isobaric conditions of testing (grey segments) and the stabilisation period between two consecutive isobars (green segments, shaded areas). The temperature is pictured as dashed lines, showing the heating (red segments) during testing and the cooling (blue segments) during the stabilisation period. For the stabilisation period prior to the 5 MPa isobaric experiments, the pressure was dropped to 3.5 MPa to allow the sample to fill the whole measurement cavity.

**Figure 2 polymers-17-03086-f002:**
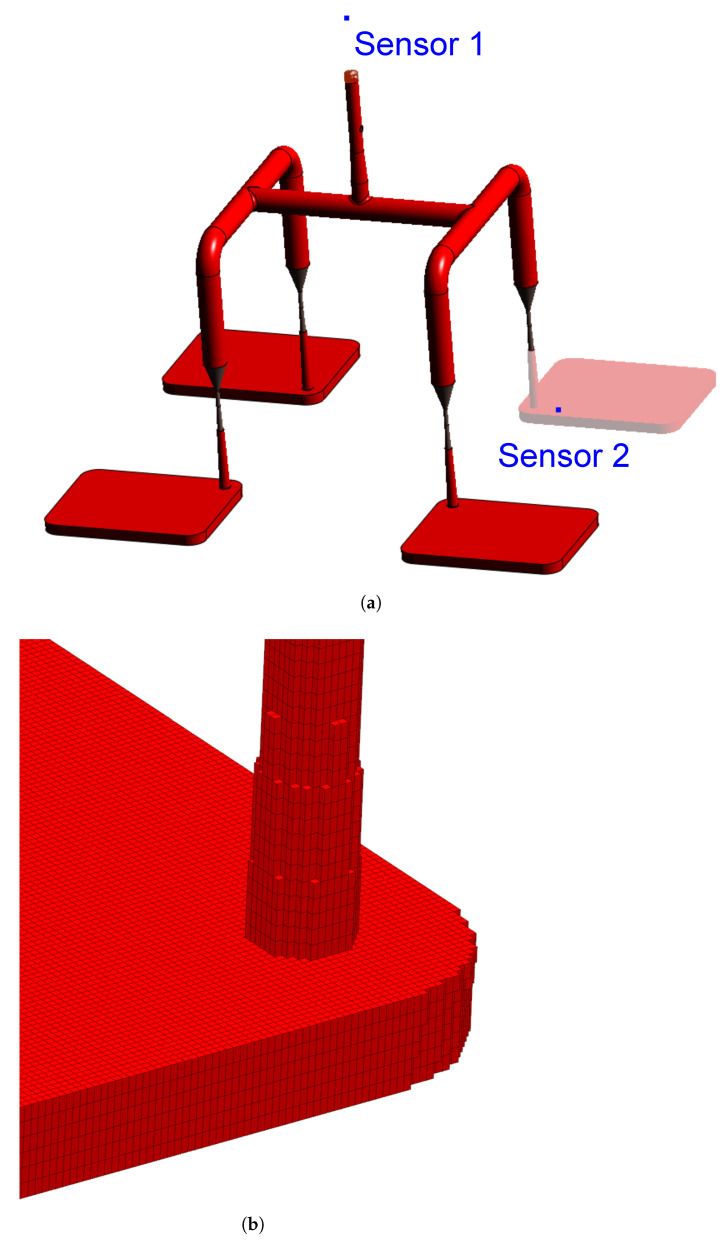
Runner system and cavity geometry applied in the simulation, showing the position of the pressure and temperature sensors (**a**), where sensor 1 is located in the mould inlet and sensor 2 is placed inside the cavity facing the flow entrance into the cavity; and details of the mesh (**b**) are employed in the volume discretisation. The geometry is inspired by the work of Traintinger [30].

**Figure 3 polymers-17-03086-f003:**
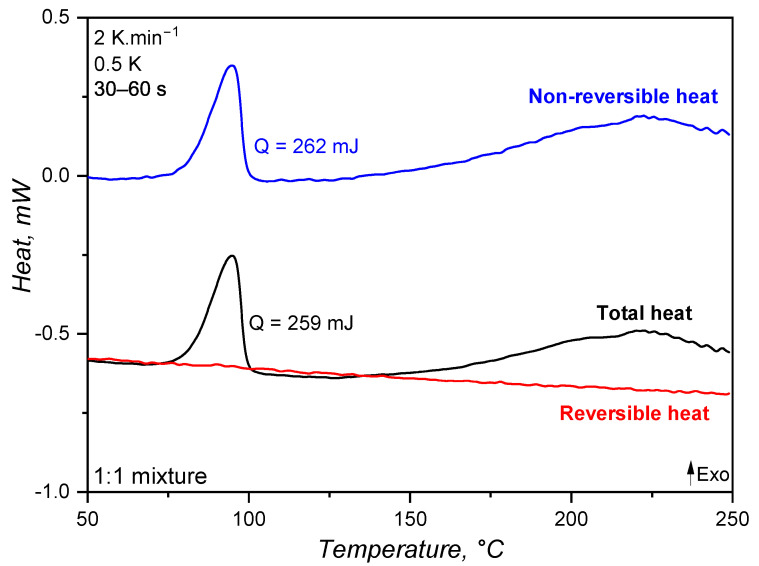
Modulated temperature DSC thermogram differentiating the total, the non-reversible, and the reversible (employed to determine $c_{p}$) heat quantities for the 1:1 LSR mixture. Sample mass is 15–20 mg. The coefficient of variation for the measurement is 7%.

**Figure 4 polymers-17-03086-f004:**
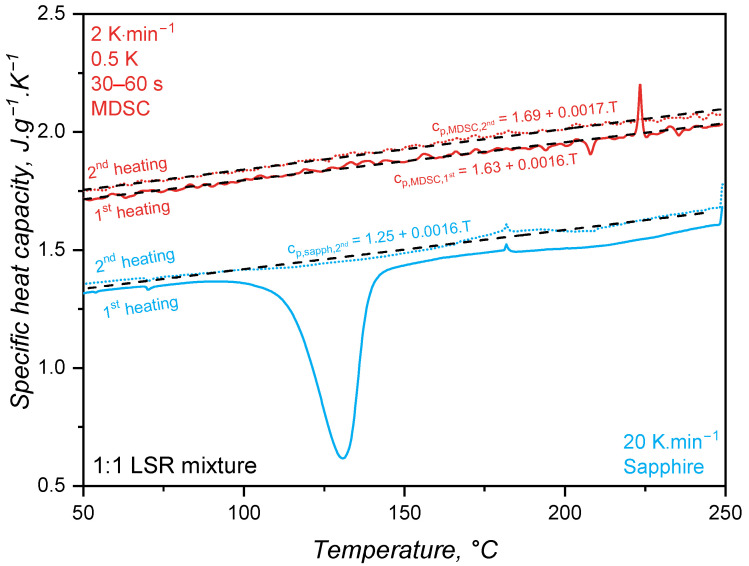
Comparison of the LSR’s specific heat capacity values determined employing the sapphire (blue lines) and the MDSC (red lines) methods. The full lines represent the first heat run, while the sample is still uncured until the crosslinking reaction starts, whereas the dotted lines show the second heat run for the fully crosslinked samples. The dashed lines are linear fittings (equations are in the plot) with $R_{adj}^{2}>0.99$. The coefficient of variation for the measurement is 7%.

**Figure 5 polymers-17-03086-f005:**
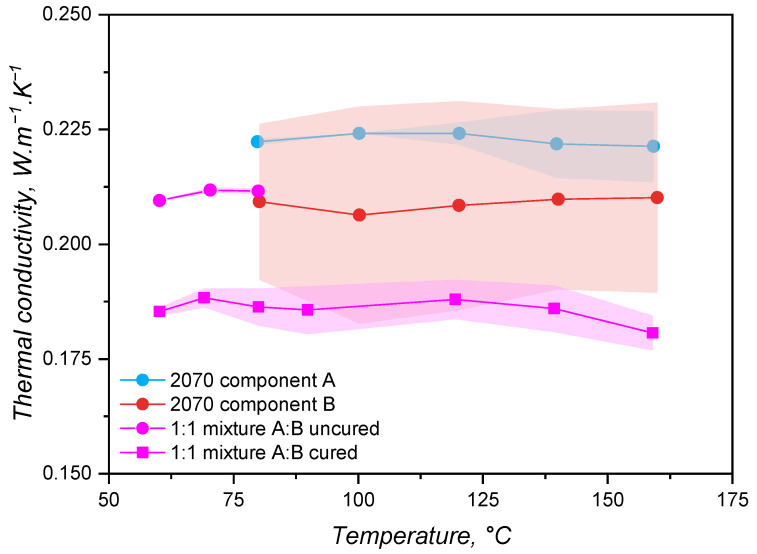
Thermal conductivity variation with temperature for the individual A and B components, as well as for the uncured and cured mixtures. The components and the uncured mixture were analysed via a transient method, while the cured mixture was investigated by employing a steady-state method. The symbols indicate the average of three measurements and the shaded areas represent the standard deviation.

**Figure 6 polymers-17-03086-f006:**
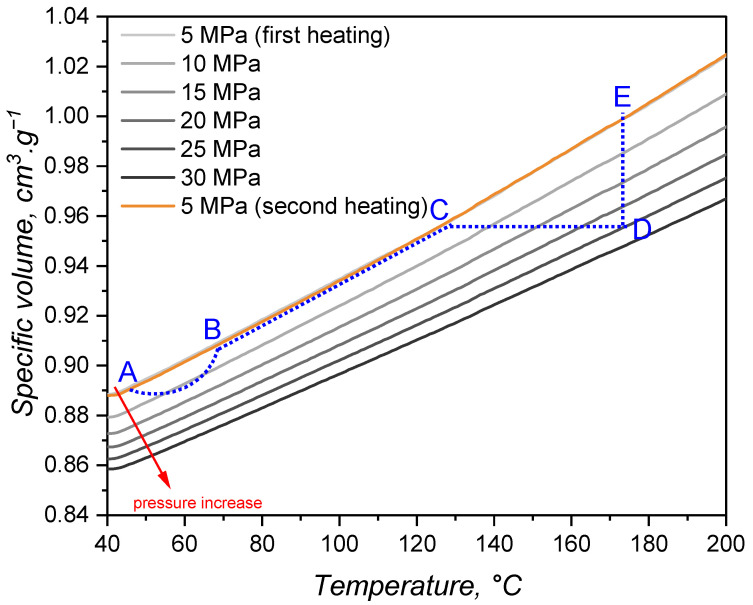
Variation in the specific volume (one measurement) as a function of pressure and temperature. The orange line represents the second measurement at 5 MPa at which the sample is fully crosslinked and is on top of the light gray line representing the 5 MPa isobar. The dotted blue line represents a hypothetical injection moulding cycle, with the following phases: A-B, injection; B-C, expansion; C-D, compression; D-E, demoulding; and E-A, cooling.

**Figure 7 polymers-17-03086-f007:**
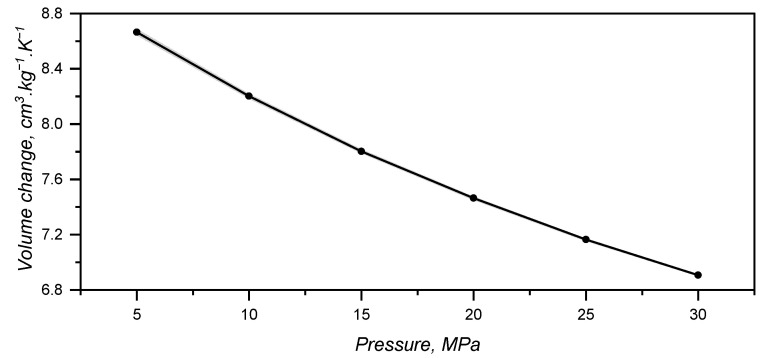
Rate of volume change due to temperature (cm^3^·kg^−1^·K^−1^) for each tested pressure. The shaded area around the plot represents the error connected to the linear fitting of the data in Figure 6. The coefficients of determination for such linear fittings are higher than 0.999.

**Figure 8 polymers-17-03086-f008:**
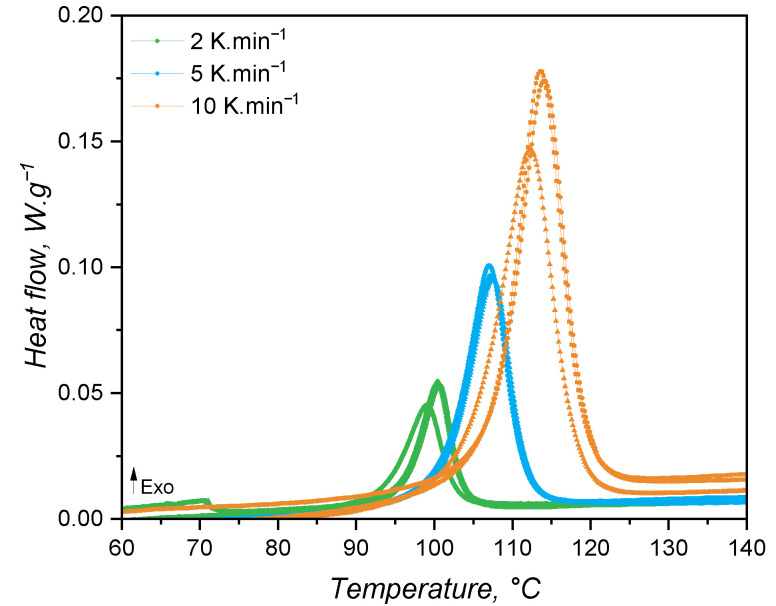
Dynamic scanning calorimetry analysis of LSR under three heating rates and inert atmosphere, showing the exothermic thermal event related to crosslinking. Each symbol/line represents one repetition.

**Figure 9 polymers-17-03086-f009:**
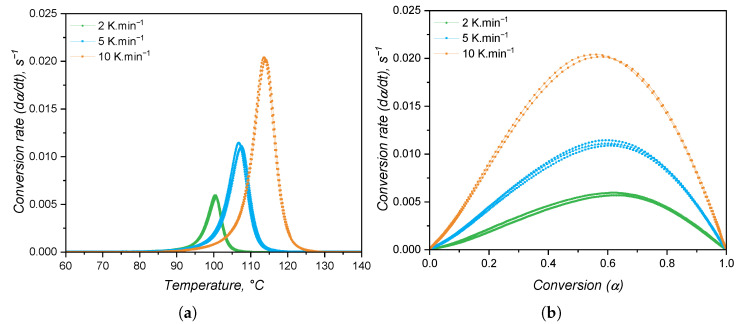
Conversion rate dα/dt for LSR at several heating rates as a function of temperature (**a**) and conversion (**b**).

**Figure 10 polymers-17-03086-f010:**
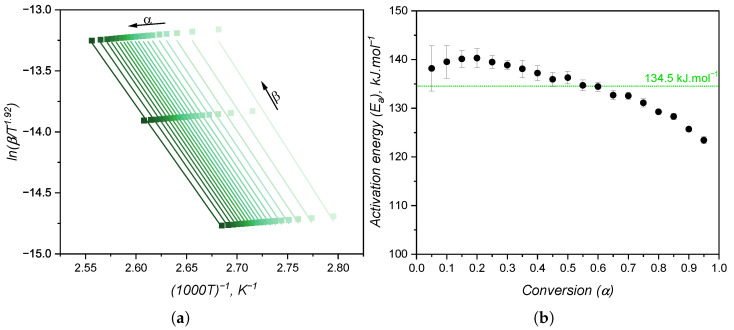
Friedman-like isoconversional approach for calculating the activation energy (**a**) and the activation energy as a function of conversion for the calorimetric approach (**b**).

**Figure 11 polymers-17-03086-f011:**
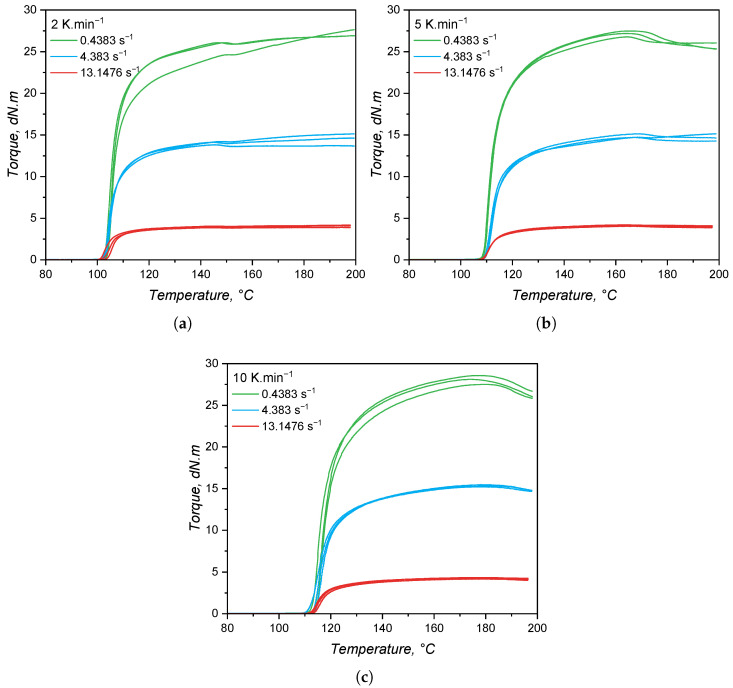
Variation in torque over temperature during the experiments at the rubber process analyser (RPA) for various shear rates at 2 K·min^−1^ (**a**), 5 K·min^−1^ (**b**), and 10 K·min^−1^ (**c**).

**Figure 12 polymers-17-03086-f012:**
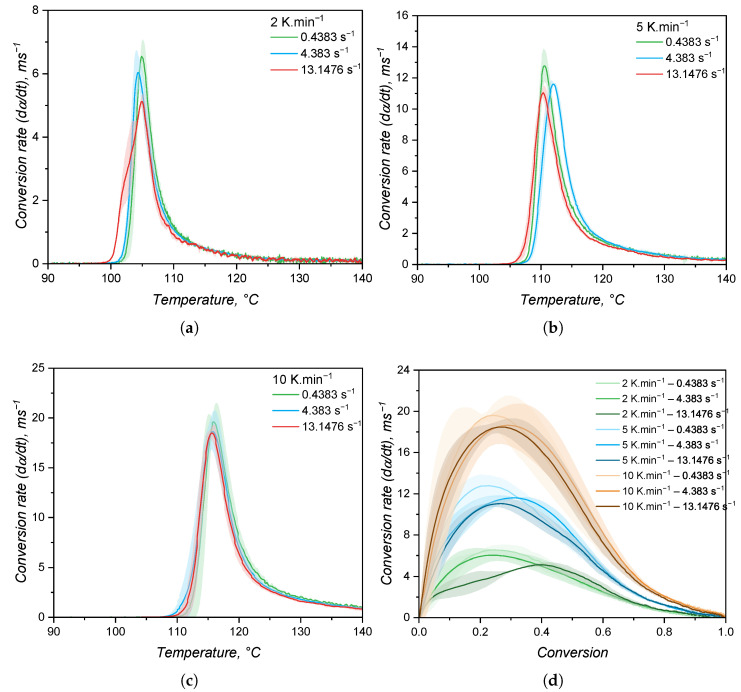
Conversion rate dα/dt measured via RPA for LSR as a function of temperature at various shear rates at 2 K·min^−1^ (**a**), 5 K·min^−1^ (**b**), and 10 K·min^−1^ (**c**). The variation in dα/dt over conversion is shown in (**d**). The shaded areas around the data lines (average) represent the standard deviation around the average.

**Figure 13 polymers-17-03086-f013:**
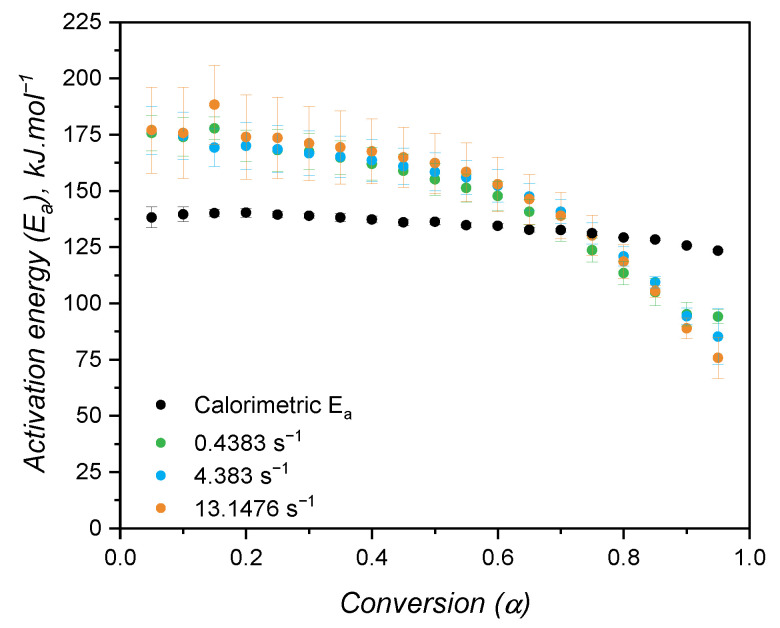
Activation energy values for the rheological approach (coloured dots) compared to the calorimetric method (black dots).

**Figure 14 polymers-17-03086-f014:**
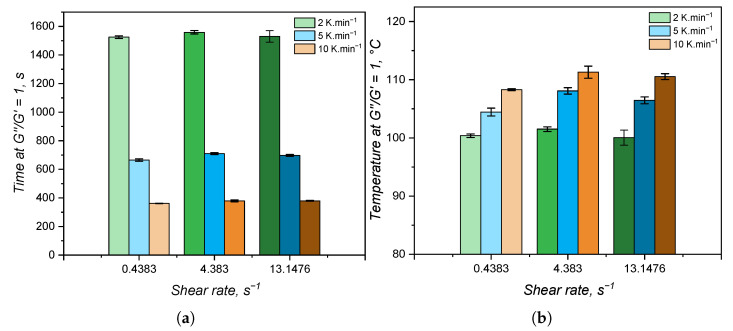
Gel time (**a**) and temperature (**b**) for all heating rates and shear rates as studied by the rheology-based approach.

**Figure 15 polymers-17-03086-f015:**
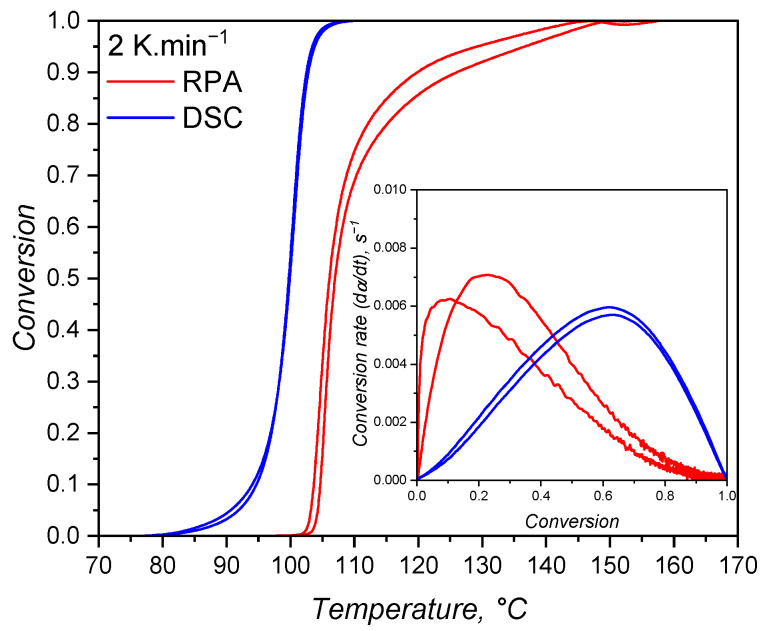
Comparison between the calculated conversion for the DSC and the RPA (0.4383 s^−1^) approaches at 2 K·min^−1^ for two replicates. In the details of the plot, the conversion rate as a function of the conversion is compared.

**Figure 16 polymers-17-03086-f016:**
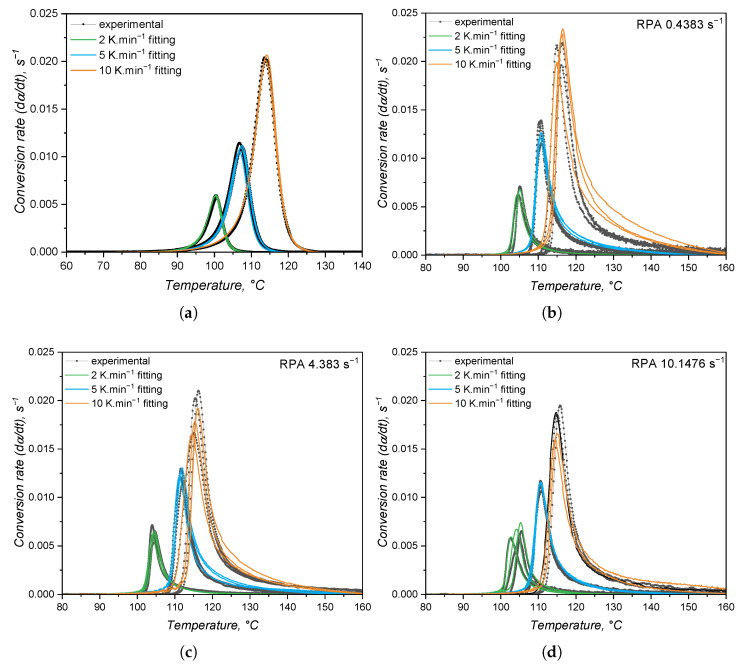
Comparison of experimental data (black symbols) and fitting according to the Kamal model (coloured lines) for the conversion rate dα/dt at several heating rates as a function of temperature for the DSC experiments (**a**) and for the RPA measurements (**b**–**d**) at different shear rates.

**Figure 17 polymers-17-03086-f017:**
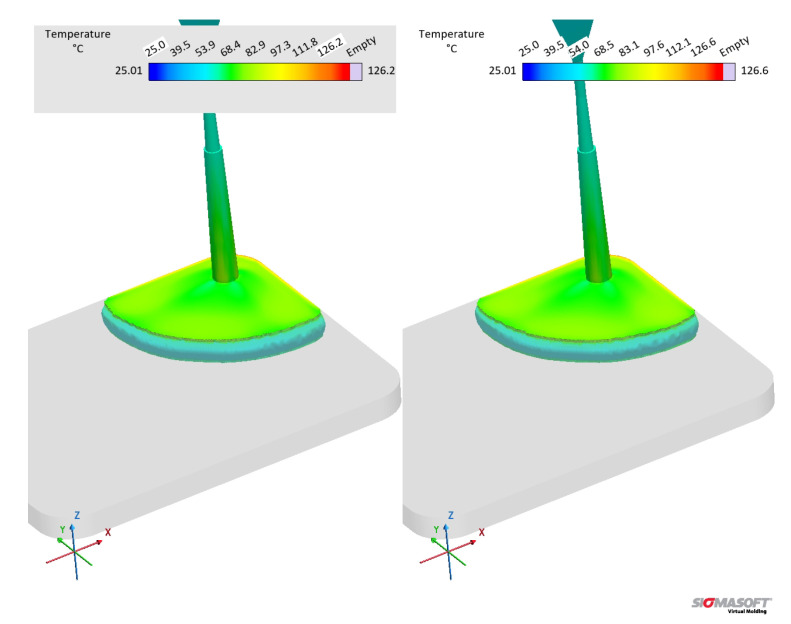
Simulated LSR flow behaviour for datasets A (**left**, LAOS) and B (**right**, HPCR) at 20% filling for one cavity.

**Figure 18 polymers-17-03086-f018:**
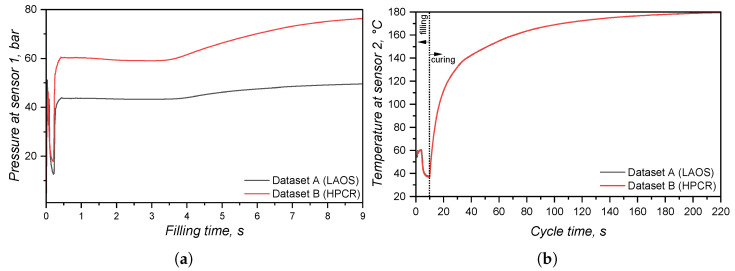
Simulated pressure at the sprue (sensor 1) during the filling stage (**a**) and simulated temperature at one cavity (sensor 2) during the injection moulding cycle (**b**) for two different datasets with distinct viscosity input, as described in Table 4.

**Figure 19 polymers-17-03086-f019:**
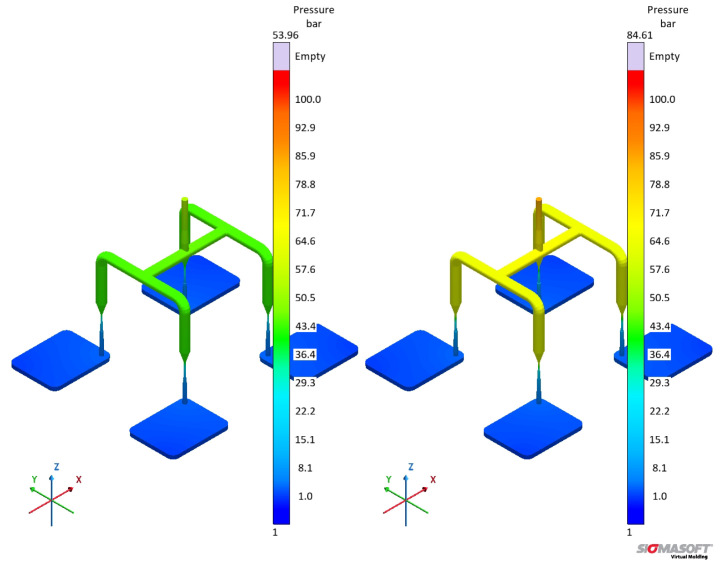
Simulated pressure for the whole studied volume, including the sprue and the running system, for datasets A (**left**, LAOS) and B (**right**, HPCR), where the pressure difference is evident at the sprue at the end of filling.

**Figure 20 polymers-17-03086-f020:**
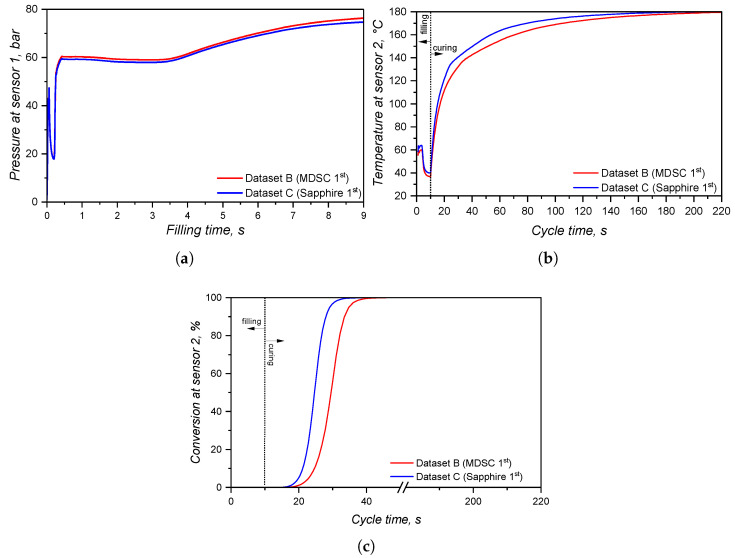
Simulated pressure at the sprue (sensor 1) during the filling stage (**a**), simulated temperature at one cavity (sensor 2) during the injection moulding cycle (**b**), and curing degree (**c**) for the two different datasets, B and C, with distinct specific heat capacity input, as shown in Table 1. The $c_{P}$ data concerning MDSC first and sapphire first can be consulted in Figure 4.

**Figure 21 polymers-17-03086-f021:**
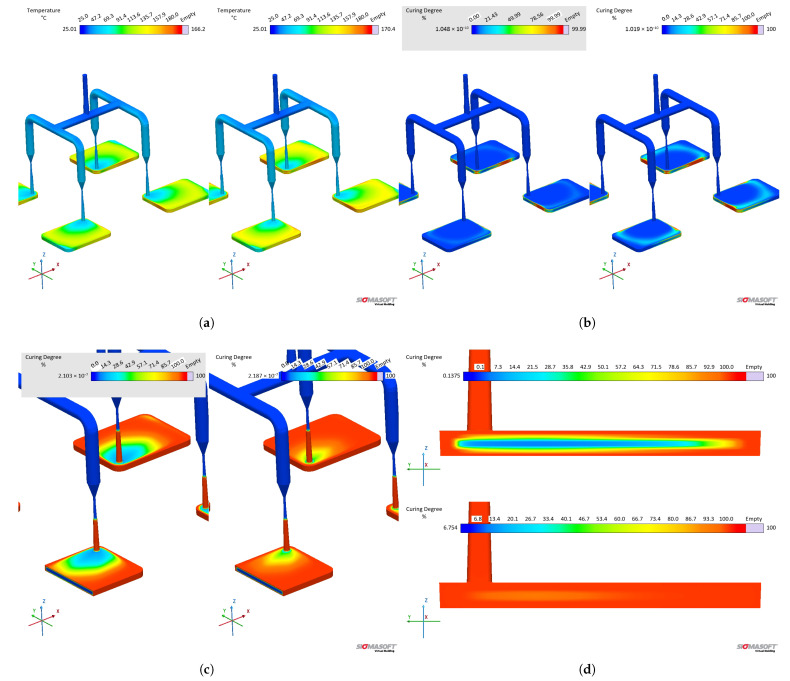
Simulated cavity temperature at the end of the filling stage for datasets B (**left**, MDSC first) and C (**right**, sapphire first) (**a**), with the correspondent curing degree (conversion 0–100%) (**b**). Curing degree at t = 25 s (**c**) and at t = 55 s (**d**) for datasets B (**left** and **top**, MDSC first) and C (**right** and **bottom**, sapphire first).

**Figure 22 polymers-17-03086-f022:**
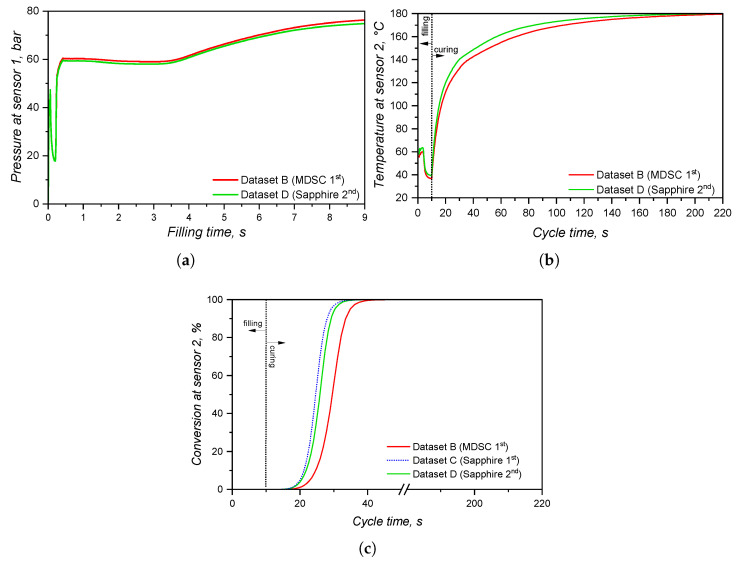
Simulated pressure at the sprue (sensor 1) during the filling stage (**a**), simulated temperature at one cavity (sensor 2) during the injection moulding cycle (**b**), and curing degree (**c**) for the different datasets B and D with distinct specific heat capacity input, as shown in Table 1. The$c_{P}$ data concerning MDSC first and sapphire first can be consulted in Figure 4. The conversion values for dataset C are shown in (**c**) for comparison purposes.

**Figure 23 polymers-17-03086-f023:**
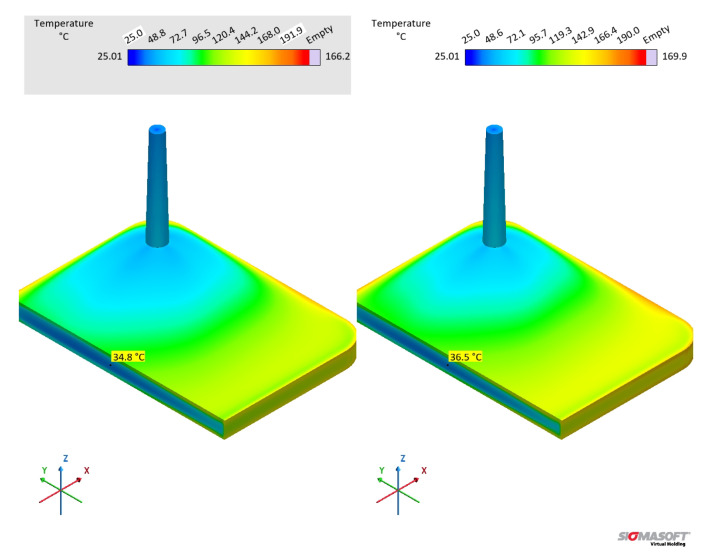
Simulated temperature at the cavity for datasets B (**left**, MDSC first) and D (**right**, sapphire second) at the end of the filling step, with the assigned local temperature at the centre of the cavity cross-section: 34.8 °C (**left**) and 36.5 °C (**right**).

**Figure 24 polymers-17-03086-f024:**
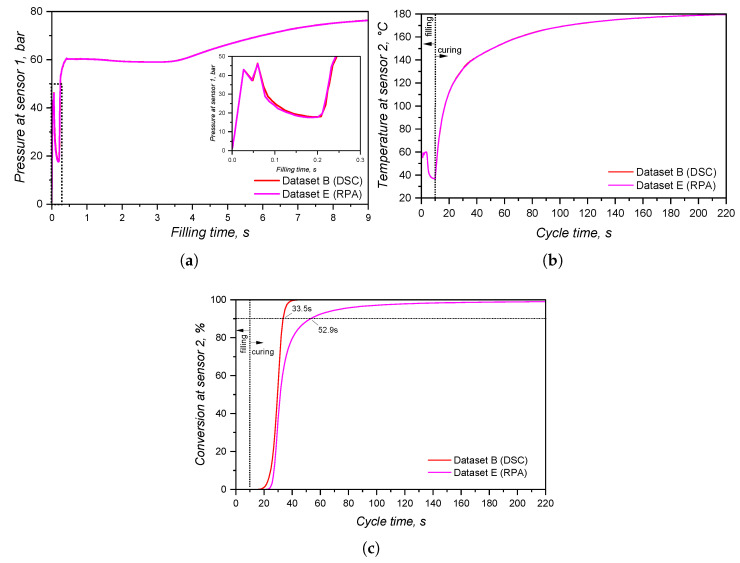
Simulated pressure at the sprue (sensor 1) during the filling stage (**a**), simulated temperature at one cavity (sensor 2) during the injection moulding cycle (**b**), and curing degree (**c**) for the different datasets B and E with distinct curing kinetics inputs as shown in Table 6. The dashed area shown in (**a**) between 0 and 0.3 s is enlarged in the same plot to aid differentiation of the curves. The time at which the curing degree = 90% is marked in (**c**) as 33.5 s for dataset B and 52.9 s for dataset E.

**Figure 25 polymers-17-03086-f025:**
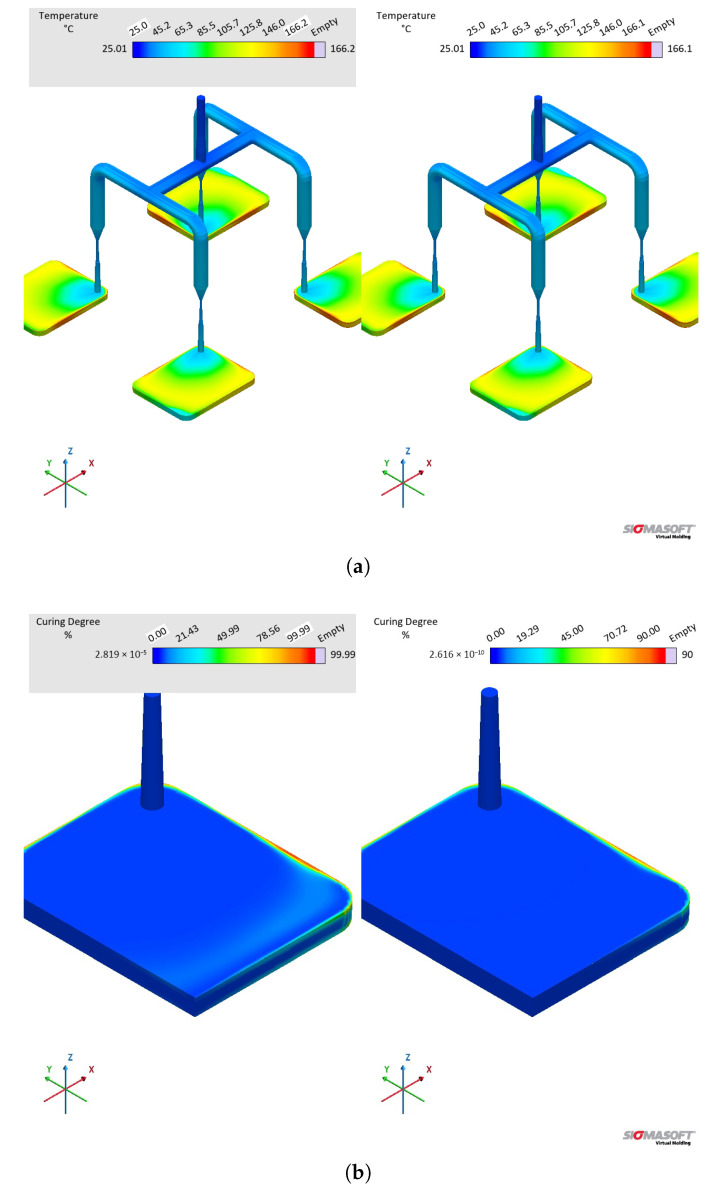
Simulated cavity temperature at the end of the filling stage for datasets B (**left**, DSC) and E (**right**, RPA) (**a**), with the correspondent curing degree (conversion 0–100%) (**b**).

**Figure 26 polymers-17-03086-f026:**
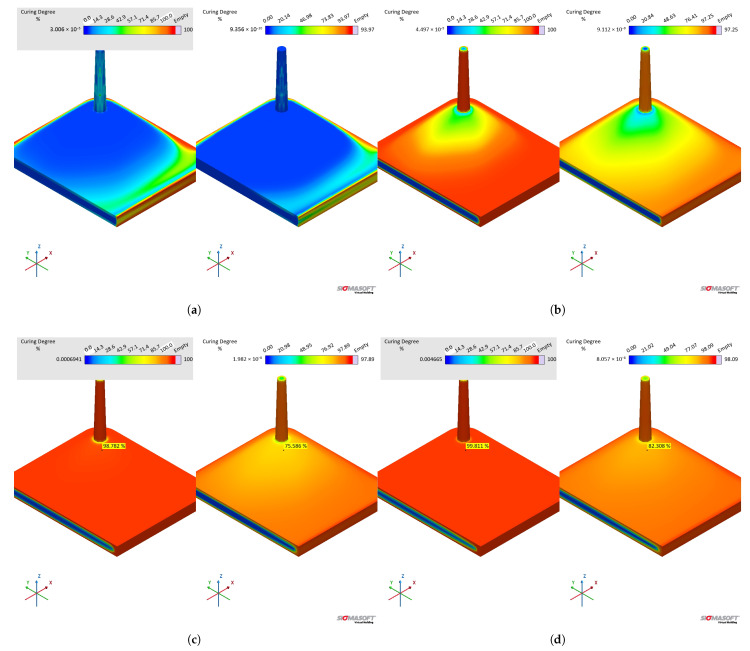
Simulated cavity curing degree (conversion 0–100%) for datasets B (**left**, DSC) and E (**right**, RPA) at the cycle times t = 15 s (**a**), 30 s (**b**), 40 s (**c**), and 45 s (**d**). The local surface indications for the curing degree assign 98.78% (**left**) and 85.59% (**right**) at (**c**); and 98.81% (**left**) and 82.31% (**right**) at (**d**).

**Table 1 polymers-17-03086-t001:** Material data employed to run the comparison simulation routines according to each sub-dataset and the employed characterisation technique. DSC* denotes calorimetry data without input for the curing enthalpy; first and second denote the runs sequences performed in the DSC analyses.

Dataset	A	B	C	D	E
Viscosity	LAOS	HPCR	HPCR	HPCR	HPCR
$c_{p}$	MDSC first	MDSC first	sapphire first	sapphire second	MDSC first
λ			cured sample		
pvT			as determined		
Curing	DSC	DSC	DSC*	DSC	RPA

**Table 2 polymers-17-03086-t002:** Thermal conductivity averages $\bar{\lambda }$ for the whole studied temperature range and respective standard deviations σ(λ), including the associated measurement error.

Sample	$\bar{\lambda }$	σ(λ)	Error
2070 A	0.223	0.001	7%
2070 B	0.209	0.001	7%
A + B uncured	0.211	0.005	7%
A + B cured	0.186	0.002	5%

**Table 3 polymers-17-03086-t003:** Kinetic parameters (average) determined after fitting of the experimental conversion rate dα/dt to the Kamal model (Equation (Equation 14)) for LSR employing various experimental procedures (DSC and rheological, from which the employed shear rates are shown).

Parameter	DSC	0.4383 s^−1^	4.383 s^−1^	13.1476 s^−1^
A_1_, s^−1^	$1.07\times 10^{24}$	$1.00\times 10^{5}$	$1.00\times 10^{5}$	$1.00\times 10^{5}$
A_2_, s^−1^	$4.98\times 10^{12}$	$1.35\times 10^{17}$	$1.40\times 10^{17}$	$7.24\times 10^{12}$
E_1_, kJ·mol^−1^	193.6	171.3	158.9	178.3
E_2_, kJ·mol^−1^	100.9	133.9	134.1	102.2
m	1.52	0.73	0.78	0.84
n	1.24	3.00	3.00	3.00

**Table 4 polymers-17-03086-t004:** Carreau–Yasuda model parameters as obtained by fitting the viscosity data from datasets A and B.

Parameter	A	B
$\eta_{\infty }$, Pa.s	0.0248	0.00917
$\eta_{0}$, Pa.s	22.75	8.0
*a*, -	5.0	5.0
*n*, -	0.103	0.368
λ, s	12.64	12.0
$T_{0}$, °C	72.0	72.0
$T_{S}$, °C	−273.0	−103.22

**Table 5 polymers-17-03086-t005:** Comparison of simulated sprue pressure metrics for viscosity datasets.

Dataset	Peak Sprue Pressure (bar)	Integral Pressure × Time (bar·s)
A (LAOS)	50.21	201.2
B (HPCR)	77.31	270.5

**Table 6 polymers-17-03086-t006:** Kinetic parameters (average) determined after fitting of the experimental conversion rate $\frac{d\alpha }{dt}$ to the Kamal model (Equation (Equation 14)) for LSR, employing the calorimetry approach (dataset B) and the rheological method (dataset E).

Parameter	B	E
log(A_1_), s^−1^	24.03	5.00
log(A_2_), s^−1^	12.69	17.13
E_1_, kJ·mol^−1^	193.6	171.3
E_2_, kJ·mol^−1^	100.9	133.9
m, -	1.52	0.73
n, -	1.24	3.00
Enthalpy, kJ.kg^−1^	8.15	8.15

## Data Availability

The data presented in this study are available on request from the corresponding author due to privacy.

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
