# Peer review of "How Method Matters: The Impact of Material Characterisation Techniques on Liquid Silicone Rubber Injection Moulding Simulations"

_polymers, 2025, doi:10.3390/polym17223086_

Round 1
Reviewer 1 Report
Comments and Suggestions for Authors
This paper investigates how different experimental techniques for material characterization affect the accuracy and predictive quality of injection moulding simulations for highly filled liquid silicone rubber (LSR). The authors systematically compare multiple methods for measuring viscosity (LAOS vs. HPCR), specific heat capacity (modulated DSC vs. sapphire method), thermal conductivity (transient vs. steady-state), and curing kinetics (calorimetry vs. rheology), using a single commercial LSR compound. The study builds five distinct material datasets and evaluates their impact on simulated pressure, temperature, and degree of cure in a controlled simulation setup (a four-cavity LSR mould). However, several questions need to be answered before the publishment of this manuscript:
1. The paper reports using triplicates for measurements, but it is unclear whether statistical analysis (e.g., error bars in figures, ANOVA) was used to confirm whether observed differences between methods are significant. Could the authors provide confidence intervals or standard deviations for at least key comparisons (e.g., cp curves, curing kinetics)?
2. The simulation differences are clearly analyzed, but was any validation performed against real injection moulding data (e.g., cycle time, temperature profiles, or degree of cure in actual parts)? Even partial validation (for one dataset) would help establish the relevance of simulation deviations shown.
3. Was the simulation platform (SIGMASOFT) capable of fully implementing all kinetic models, particularly the Kamal model with two activation energies from Equation 14? If not, were any simplifications made, and how might that affect the interpretation of differences between rheology- and DSC-derived kinetics?
4. Figure 4 (page 13) shows that the sapphire method introduces enthalpy-driven artifacts in cp values, while MDSC does not. However, in practice, is one more beneficial for simulation? Should cp include heat release indirectly (as with sapphire) or should latent heat be entered separately (as in MDSC + enthalpy input)?
5. With over 937,000 elements in the simulation (page 10), did the authors perform any mesh sensitivity analysis? This would be particularly relevant to ensure that differences in predictions stem from material property datasets and not from numerical artifacts.
Author Response
Please see the attachment. The revised manuscript is attached next to the comments, in the same .pdf file.

Reviewer 2 Report
Comments and Suggestions for Authors
The manuscript systematically compares material characterization routes for an LSR (Silopren 2070, ~32 wt% filler) and quantifies how those choices (viscosity, thermal conductivity, pvT, and curing kinetics) propagate into injection-molding simulation outputs. Using both sapphire and modulated DSC for, DSC and RPA for kinetics, and LAOS vs HPCR for viscosity, the authors show curing-related datasets dominate cycle-time/degree-of-cure predictions, while viscosity mainly shifts injection pressures. The main takeaway is that dataset selection, especially for (with enthalpy) and kinetics, critically governs predictive fidelity in LSR IM simulations.
Timely and useful study with strong methodological depth. The writing would benefit from tighter positioning of novelty, clearer reproducibility details for each test, and a crisper link from data-selection differences to specific simulation deltas.
- Please make explicit, in the Introduction and end of the Abstract, what is new versus prior LSR IM studies. Right now, the gap and “why this matters” are implied rather than stated plainly.
- Consolidate key properties (A: B=1:1, mixing protocol, filler content, Mw/Mn, curing temperature windows) into a single table for reproducibility. Some are scattered across Methods and prior work.
- The comparison shows DSC captures early conversion while RPA misses the onset; please quantify the fraction of the overall heat (from DSC) that occurs before the RPA “detectable” point, and discuss implications for Kamal parameter bias (m vs n dominance) and cycle-time predictions.
- You attribute pressure differences to dataset choice; include a small table with peak sprue pressure and integral (pressure×time) during filling for LAOS vs HPCR, and clarify the temperature used for the viscosity.
- Since cured vs uncured were measured by different methods (steady vs transient), please add an uncertainty budget and explicitly caution against interpreting the cured sample’s lower value as physical rather than methodological.
- Provide the exact SIGMASOFT version/build, solver settings (time step, convergence criteria). Also include the boundary coefficients you used and why.
- You discuss ejection at 75–95% Doc. Please state the chosen pass/fail criterion used for “cycle time” in your comparisons and justify it with a reference or a short note on post-ejection cure.
- Please include a brief mesh-independence study for the SIGMASOFT model.
- Consider adding a one-page symbol table.
Author Response

(The authors gave the same response as above.)

Round 2
Reviewer 2 Report
Comments and Suggestions for Authors
I appreciate the clarification regarding mesh size and the rationale for using a consistent mesh across simulations. However, I still recommend that a brief mesh-independence test be included or at least summarized in supplementary information. Such a test is relevant to all mesh-based numerical studies to ensure that the chosen discretization is sufficiently fine to support the experimental validation and that numerical artifacts do not influence the reported pressure, temperature, and degree-of-cure trends.
Author Response
Dear reviewer,
We appreciate the revision and we submitted our comments to the attached document. A revised manuscript was also uploaded.
